# Enhanced wind mixing and deepened mixed layer in the Pacific Arctic shelf seas with low summer sea ice

Yuanqi Wang [1], Zhixuan Feng [1,2] ✉, Peigen Lin[3], Hongjun Song [4], Jicai Zhang[1], Hui Wu[1,5], Haiyan Jin[2,3,6], Jianfang Chen[2,3,6], Di Qi[7] & Jacqueline M. Grebmeier[8]

The Arctic Ocean has experienced significant sea ice loss over recent decades, shifting towards a thinner and more mobile seasonal ice regime. However, the impacts of these transformations on the upper ocean dynamics of the biologically productive Pacific Arctic continental shelves remain underexplored. Here, we quantified the summer upper mixed layer depth and analyzed its interannual to decadal evolution with sea ice and atmospheric forcing, using hydrographic observations and model reanalysis from 1996 to 2021. Before 2006, a shoaling summer mixed layer was associated with sea ice loss and surface warming. After 2007, however, the upper mixed layer reversed to a generally deepening trend due to markedly lengthened open water duration, enhanced wind-induced mixing, and reduced ice meltwater input. Our findings reveal a shift in the primary drivers of upper ocean dynamics, with surface buoyancy flux dominant initially, followed by a shift to wind forcing despite continued sea ice decline. These changes in upper ocean structure and forcing mechanisms may have substantial implications for the marine ecosystem, potentially contributing to unusual fall phytoplankton blooms and intensified ocean acidification observed in the past decade.

The Pacific Arctic continental shelves, comprising the northern Bering and Chukchi seas, are among the most productive high-latitude systems[1–3] but also experiencing drastic climate changes[4]. Longer ice-free periods[5,6], more solar heating[7], and increasing Pacific water inflow[8] through the Bering Strait promote ocean warming and freshening[9,10] in this region. The state of the upper ocean is crucial to regional productivity since a strong stratification at the mixed layer base effectively isolates the convection of heat[11,12] and nutrients[13], hindering phytoplankton blooms in the euphotic zone. Several studies suggested that primary production should decrease with a freshwater lens[14,15] under rapid warming and sea ice loss. However, satellite-based estimation showed increased primary production[16,17] with greater light penetration and longer open water seasons in the Arctic[18]. The uncertainty of phytoplankton dynamics evaluated through a 20-year time series analysis specific to July as part of the Distributed Biological Observatory (DBO) in this region indicates productivity is equivocal, suggesting variability in sea ice regimes influences water column structure, nutrient cycling, and productivity seasonally[19].

[1]State Key Laboratory of Estuarine and Coastal Research, School of Marine Sciences, and Institute of Eco-Chongming, East China Normal University, Shanghai, China. [2]State Key Laboratory of Satellite Ocean Environment Dynamics, Second Institute of Oceanography, Ministry of Natural Resources, Hangzhou, China. [3]School of Oceanography, Shanghai Jiao Tong University, Shanghai, China. [4]Observation and Research Station of Bohai Strait Eco-Corridor, First Institute of Oceanography, Ministry of Natural Resources, Qingdao, China. [5]School of Mathematical Sciences, and Key Laboratory of MEA, Ministry of Education, East China Normal University, Shanghai, China. [6]Key Laboratory of Marine Ecosystem Dynamics, Second Institute of Oceanography, Ministry of Natural Resources, Hangzhou, China. [7]Polar and Marine Research Institute, Jimei University, Xiamen, China. [8]Chesapeake Biological Laboratory, University of Maryland Center for Environmental Science, Solomons, MD, USA. ✉e-mail: zxfeng@sklec.ecnu.edu.cn

The northern Bering and Chukchi seas have multiple benthic biological hotspots[20] (Supplementary Fig. 1a) and support a variety of upper trophic level species[21]. Earlier sea ice retreat weakens the contribution of ice algae to benthic food and promotes the region to become a more pelagic-dominated marine ecosystem[22,23]. The open water duration and upper mixed layer may greatly impact the exchange of heat, gas, nutrients, biological production, and the benthic communities on the shelf. In recent years, rapid climate change may have induced a northward shift of macrofaunal in this region[3,4,20,24,25]. Previous studies suggested that the mixed layer depth (MLD) in the western Arctic, especially in the Canada Basin, showed a shoaling trend from the 1970s to 2000s[26–28]. At the same time, cruise observations[29,30] on a short time scale revealed that wind-induced mixing was enhanced and the mixed layer deepened in an increasingly ice-free Arctic Ocean[31]. Because the Pacific Arctic has now transitioned to a new normal with much less sea ice than 2–3 decades ago[32], this leads to an open question about whether ocean dynamics may have shifted in recent years. Delineating the upper ocean dynamics on these highly productive shelves during the growing season is fundamental for understanding ongoing ecological changes under rapid sea ice decline.

Here, we analyze 23,320 shipboard CTD/XCTD profiles and model reanalysis results from 1996 to 2021 to depict the interannual variations and decadal trends of the upper ocean properties on the Pacific Arctic shelf seas, defined as depths of 15–500 m. Our results suggest a potential regime shift in the upper ocean dynamics: a reverse trend of MLD from shoaling to deepening after 2007, a well-known year with the second lowest summer sea ice coverage since the satellite monitoring era. This finding demonstrates another robust mechanism and consequence of sea ice loss on the upper ocean dynamics, in addition to a traditional understanding of shoaling mixed layers mainly due to more ice meltwater[33–35]. We further quantify the roles of wind mixing, atmospheric buoyancy flux, and sea-ice-derived meltwater for summer MLD development and variations. Our hypothesis is that in increasingly ice-free Pacific Arctic shelf seas with prolonged open water duration, wind forcing can sufficiently mix the upper water column and become a more dominant factor than atmospheric buoyancy and ice meltwater in determining the summer mixed layer.

## Results

### Sea ice regime shift

The sea ice in the Pacific Arctic shelf seas often started retreating at the end of March and completely melted or reached the annual minimum extent in September[6,36,37]. After 2007, this region became almost ice-free in August (Fig. 1a and b). The period from 1996 to 2006 could be considered as a transition with shrinking perennial ice cover in the Pacific Arctic[5,38,39], and a new normal of low summer sea ice generally stabilized after 2007[40–42]. Most areas of the northern Bering and Chukchi shelves did not show an early ice retreat or prolonged open water duration from 1996 to 2006 (Supplementary Fig. 2a and b). The year 2007 was a potential tipping point of ice regime shift, triggered by the transition of a negative Arctic Dipole index to a positive one[43]. The increasingly positive Arctic Dipole index during 2007–2021 drove an enhanced anomalous oceanic heat flux and amplified ice–albedo feedback in the Pacific Arctic[44,45]. Besides, the negative trend of Artic Oscillation on the decadal time scale was also associated with increasing ice losses[46,47]. The Arctic Amplification[48] driven by atmospheric circulation patterns, caused the seasonal sea ice to decline further over the long term without any sign of recovery[49]. The trigger time of ice retreat (i.e., day of a year when sea ice concentration drops below 15% for three consecutive days) advanced by a mean rate of −2.6 days yr$^{-1}$ ($p < 0.1$) in the Bering Sea and −3.0 days yr$^{-1}$ ($p < 0.1$) in the Chukchi Sea in 2007–2021 (Supplementary Fig. 2c). The open water duration (i.e., periods with sea ice concentration below 15%) increased significantly with a mean rate of +2.8 days yr$^{-1}$ ($p < 0.1$) in most areas (Supplementary Fig. 2d).

The states and dynamics of the upper water column might have been altered with continuous sea ice decline and longer ice-free days. We compiled 23,320 observed CTD/XCTD profiles from various sources (Supplementary Tables 1 and 2) in 1996–2021 to quantify the upper mixed layer depth in the Pacific Arctic shelves (see the "Methods" section for MLD definition; see Supplementary Fig. 1b for regional examples of potential density profiles and MLDs). Although no significant correlation was found between observed MLD and sea ice concentration, MLD was closely related to the trigger time of ice retreat (Fig. 1c). We defined the time difference between hydrographic sampling and ice retreat day in the corresponding year as $\Delta t$ and retained 19,742 profiles with positive $\Delta t$ values (i.e., hydrographic profiles were sampled after sea ice retreat) to calculate their correlations with MLDs in each grid cell (see the "Methods" section). The time difference $\Delta t$ was positively correlated with the summer MLD in most of the Bering and Chukchi continental shelves (Fig. 1c). In other words, the closer hydrographic profiling to the annual ice retreat day, the shallower mixed layer with more content of sea-ice-derived meltwater. Theoretically, the mixed layer would deepen under sufficiently long wind mixing, and ice meltwater became less and less influential after sea ice retreat. Notably, a negative correlation was found in the subarctic southeastern Bering shelf, indicating that the physical mechanism that controls upper ocean dynamics there could be different from other regions of the Pacific Arctic.

### Reversal of mixed layer depth trends

To assess the impacts of changing sea ice and atmospheric conditions on the upper ocean, we further calculated trends of the mean summer (June–September) MLDs using a total of 18,997 quality-controlled hydrographic profiles over two separate periods: 1996–2006 versus 2007–2021 (sampling years and locations refer to Supplementary Fig. 3). There were 2835 summertime profiles in the Bering shelf during 1996–2006 and 6554 profiles after 2007. The Chukchi shelf contained 2146 profiles in 1996–2006 and 7462 in 2007–2021, respectively. The time division was based on the cold/warm transformation[50] in the Pacific Arctic and sea ice regime shifts analyzed above. Despite the sampling bias and data gaps of field observations in space and time scales, the consistency of MLD shoaling/deepening tends of sparse grid cells from the Bering to Chukchi shelves was notable (Fig. 2a and b). MLDs had a similar shoaling tendency in the southeastern Bering shelf during both periods, whereas higher-latitude regions showed exactly opposite trends in specific grid cells (Fig. 2a and b). The shoaling trends of MLDs on the northern Bering (−2.4 m yr$^{-1}$ of 2 grid cells, $p < 0.1$; Fig. 2a) and the Chukchi shelves (−1.1 m yr$^{-1}$ of 5 grid cells, $p < 0.1$) before 2007 agreed with previous studies[27,28]. However, after 2007, the mixed layer on the northern Bering and Chukchi shelves showed a remarkable deepening trend (up to 1.3 m yr$^{-1}$, $p < 0.1$), illustrated by reddish grid cells (Fig. 2b). A freshening trend was observed in some areas of the southeastern Bering shelf (−0.1 psu yr$^{-1}$, $p < 0.1$) before 2007, and then the water in the region gradually became saltier (up to 0.08 psu yr$^{-1}$, $p < 0.1$) (Supplementary Fig. 4a and c). The mixed layer temperature had a noticeable warming trend (0.6 °C yr$^{-1}$, $p < 0.1$) on the southeastern Bering shelf from 1996 to 2006, which extended to the whole middle Bering shelf (0.4 °C yr$^{-1}$, $p < 0.1$) over 2007–2021 (Supplementary Fig. 5a and c). In contrast, the mixed layer temperature and salinity on the northern Bering and Chukchi shelves did not show significant trends in both periods.

To confirm the trends revealed by unevenly distributed hydrographic profiles, we also analyzed daily temperature and salinity output from the Global Ocean Reanalysis and Simulations (GLORYS2V4) and evaluated the mixed layer properties using identical methods (Supplementary Fig. 1c). The reverse MLD trends on the northern Bering and Chukchi shelves during the two periods were broadly consistent with the observations (Fig. 2 and Supplementary Figs. 6a and 7a). The mixed layer shoaled at a mean rate of −0.47 m yr$^{-1}$

 

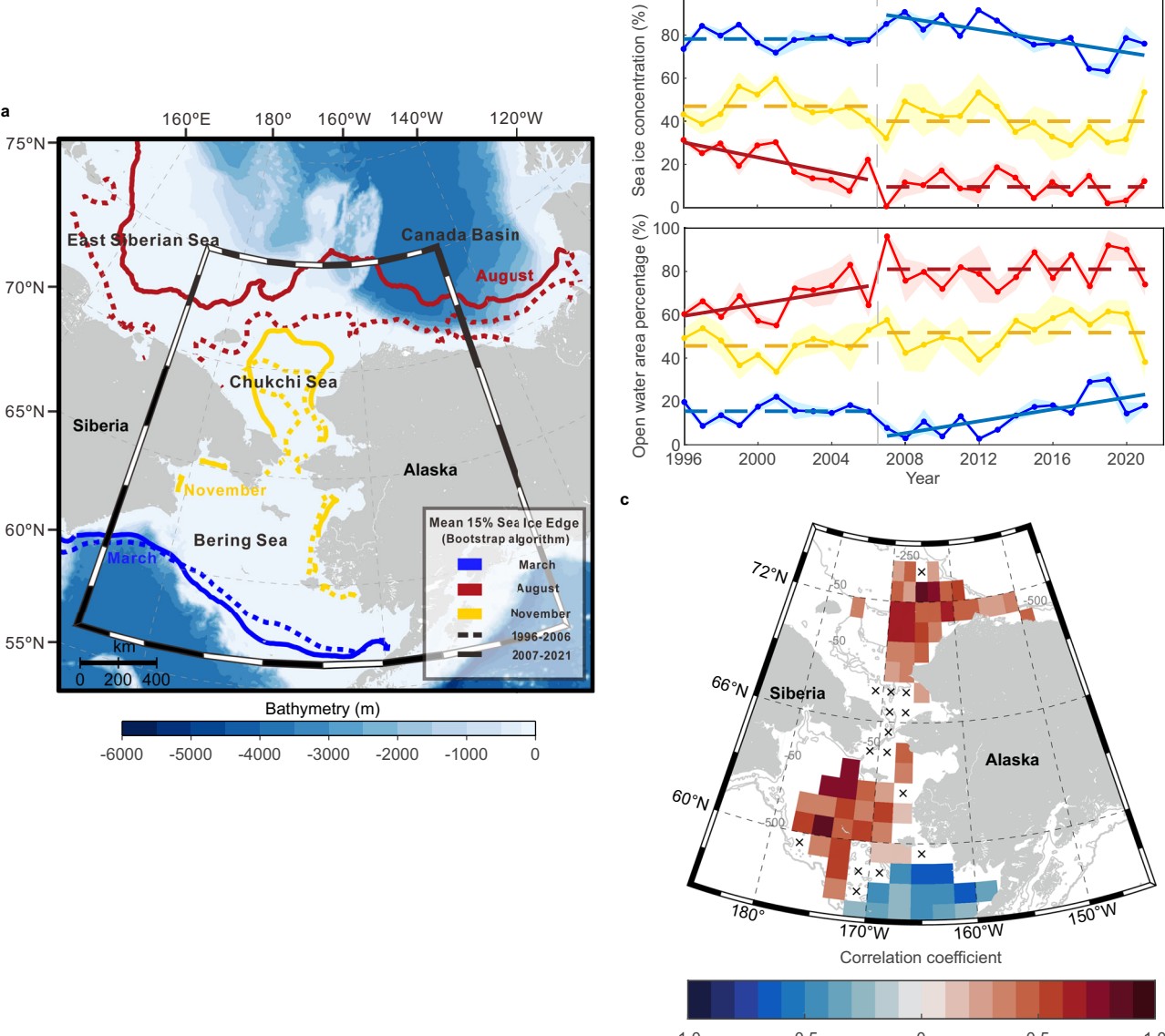

**Fig. 1 | Sea ice condition and its correlation with the observed mixed layer depth (MLD) in the Pacific Arctic shelves. a** Mean sea ice edges (sea ice concentration threshold of 15%) in mid-March (blue), mid-August (red), and mid-November (yellow) during 1996–2006 (dashed line) and 2007–2021 (solid line) in the northern Bering and Chukchi continental shelves. **b** Sea ice concentration and open water area percentage averaged over the study region from 1996 to 2021. Mean values estimated in March, August, and November are shown by blue, red, and yellow lines, respectively, and corresponding shades denote the standard error. Solid sloping lines indicate significant linear trends ($p < 0.1$) in 1996–2006 or 2007–2021, whereas the dashed lines indicate no significant trends. **c** Correlations between $\Delta t$ (i.e., the sampling time of onboard hydrographic profiles minus annual ice-retreat time) and observed MLD in the open water in 1996–2021 (see the "Methods" section). Only significant correlations ($p < 0.1$) are shown in blue (negative values) to red (positive values) colors, and each 1° × 2° grid cell has at least 20 hydrographic profiles. Black crosses illustrate the grid cells without significant correlations, and gray lines represent isobaths of 50, 250, and 500 m. Source data are provided as a Source Data file.

($p < 0.1$) in 1996–2006 and then deepened at a mean rate of 0.33 m yr$^{-1}$ ($p < 0.1$) after 2007. Due to the randomness of ship-based observations and model errors, linear trends of the MLDs based on GLORYS2V4 were more moderate than observations. Lacking tidal mixing processes in the GLORYS2V4 might result in its poor performance on the Bering shelf (Supplementary Figs. 6b and 7b). Observation datasets revealed that the inner southeastern Bering shelf (<50 m) had an overall shoaling trend (Fig. 2b). In comparison, GLORYS2V4 model results showed no obvious trends in the same region (Fig. 2d). For the high-latitude Chukchi Sea, the summer MLD trends derived from observations were similar to the model results.

## Roles of surface buoyancy flux versus wind mixing

Mechanistic understandings of surface buoyancy flux versus wind mixing are essential to disentangling upper ocean dynamics in the rapidly changing Pacific Arctic shelf seas. In principle, negative air–sea buoyancy fluxes, or net atmosphere-to-ocean heat and freshwater fluxes, make the surface ocean more stratified and relatively stable (Supplementary Fig. 8a). We calculated the air–sea heat and freshwater fluxes using ECMWF Reanalysis v5 (ERA5) data. Compared to freshwater flux (Fig. 3c), heat flux was a predominant factor in stratifying the upper ocean during 1996–2006 (Fig. 3a), while air–sea freshwater flux was stable during both periods (Fig. 3d).

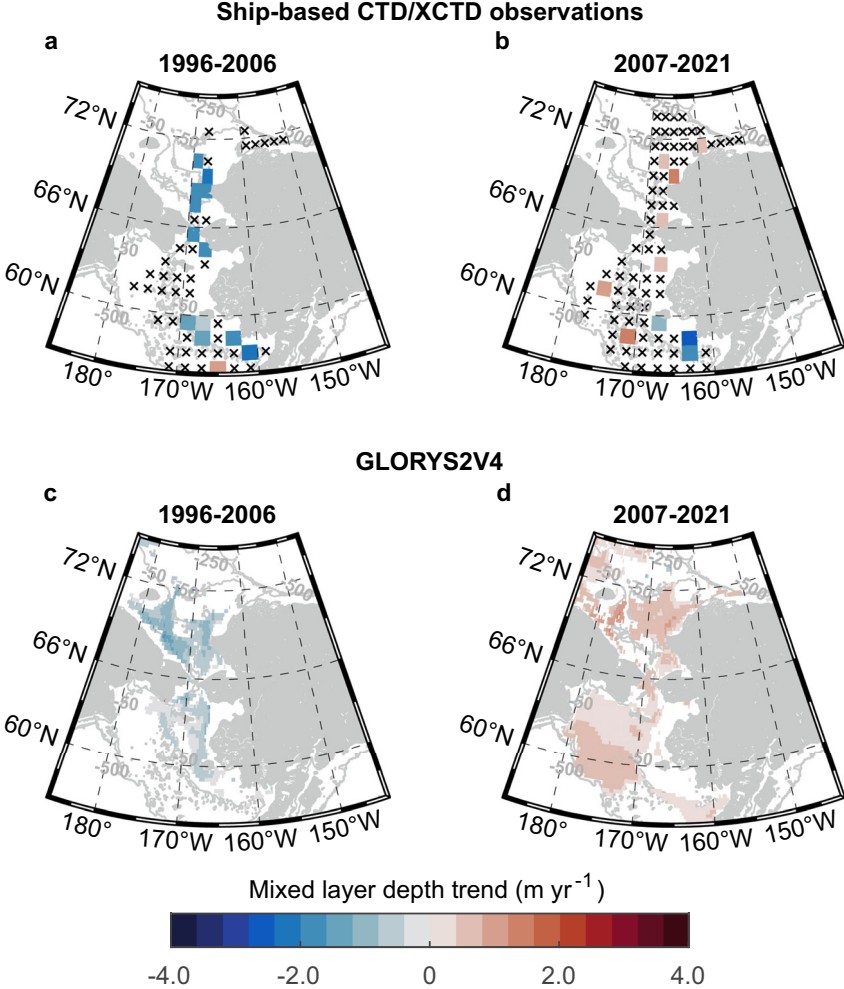

**Fig. 2 | Linear trends of summer (June–September) mixed layer depths (MLDs) over two different periods.** The trends based on ship-based CTD/XCTD profiles during **a** 1996–2006 and **b** 2007–2021 in each 1° × 2° grid cell. Statistically significant trends ($p < 0.1$) based on Global Ocean Reanalysis and Simulations (GLORYS2V4) during **c** 1996–2006 and **d** 2007–2021 in each 0.25° × 0.25° grid cell. Only significant trends with $p < 0.1$ are shown in blue (negative values) to red (positive values) colors; black crosses illustrate grid cells without significant trends. Gray lines represent isobaths of 50, 250, and 500 m. Source data are provided as a Source Data file.

Local winds over the Pacific Arctic could account for 30–50% of the synoptic scale variability[51–53], and explain 10–20% MLD variance[27,54]. We evaluated monthly mean wind stress from May to September that spanned ice melting and open water periods based on ERA5. Significant changes in wind stress mostly occurred in open seas or marginal ice zones for both periods (Supplementary Fig. 9). The wind stress from the northern Bering to Chukchi shelves decreased in the season of June–August during 1996–2006 with a mean rate of $-1.2 \times 10^{-3}$ Pa yr$^{-1}$. After 2007, the enhancement of wind stress started in May and lasted for nearly 4 months with a positive trend of $1.7 \times 10^{-3}$ Pa yr$^{-1}$, which agreed with prior results of climate models[55,56] and observations[57]. The strengthened wind stress was related to wind speed-up over the Pacific Arctic shelf under Arctic Amplification[58], because sea ice loss[59] and surface warming[55] decreased ocean surface roughness and atmospheric stability. The stronger pressure gradient between the Aleutian Low and Beaufort High also promoted geostrophic wind speed[60,61] with a positive Arctic Dipole regime since 2007[45].

For high-latitude oceans, sea ice effects must be considered since the sea-ice-derived meltwater is an additional surface buoyancy source, and the presence of pack ice largely hinders direct wind influence on the mixed layer. To quantify the roles of wind mixing, atmospheric buoyancy input, and local ice meltwater on the upper mixed layer during the summertime, we applied a 1D sea ice-ocean mixed layer model[62–64] to calculate wind-induced turbulent kinetic energy and surface buoyancy fluxes using daily model results of GLORYS2V4 (see the "Methods" section). We ascertained the evolution of MLDs from the ice-covered (i.e., sea ice concentration ≥ 15% and sea ice thickness > 0) to open waters. The Chukchi shelf contained complex water masses, including Alaska Coastal Water, ice meltwater, dense winter water, Bering Sea water, and their mixtures in summer[65]. Nevertheless, the timing and extent of advected ice meltwater and other water masses varied among regions and years[66]. Linear trends of turbulent kinetic energy and surface buoyancy fluxes were calculated using a 3-month average to smooth water mass advection and other short-time scale processes. Model results showed that the effects of wind-induced turbulent mixing decreased with a mean rate of $-4.8 \times 10^{-8}$ m$^3$ s$^{-3}$ yr$^{-1}$ over the northern Bering to southeastern Chukchi shelf before 2006 (Fig. 4a) but significantly increased with a mean rate of $5.9 \times 10^{-8}$ m$^3$ s$^{-3}$ yr$^{-1}$ over almost entire Pacific Arctic shelves after 2007 (Fig. 4b). The increased ice meltwater buoyancy near the mean summer sea ice edge in northern Chukchi shelf implied the melting of perennial sea ice before 2006 (Fig. 4e). Although the Bering shelf received more surface buoyancy during both periods (Fig. 4c and d),

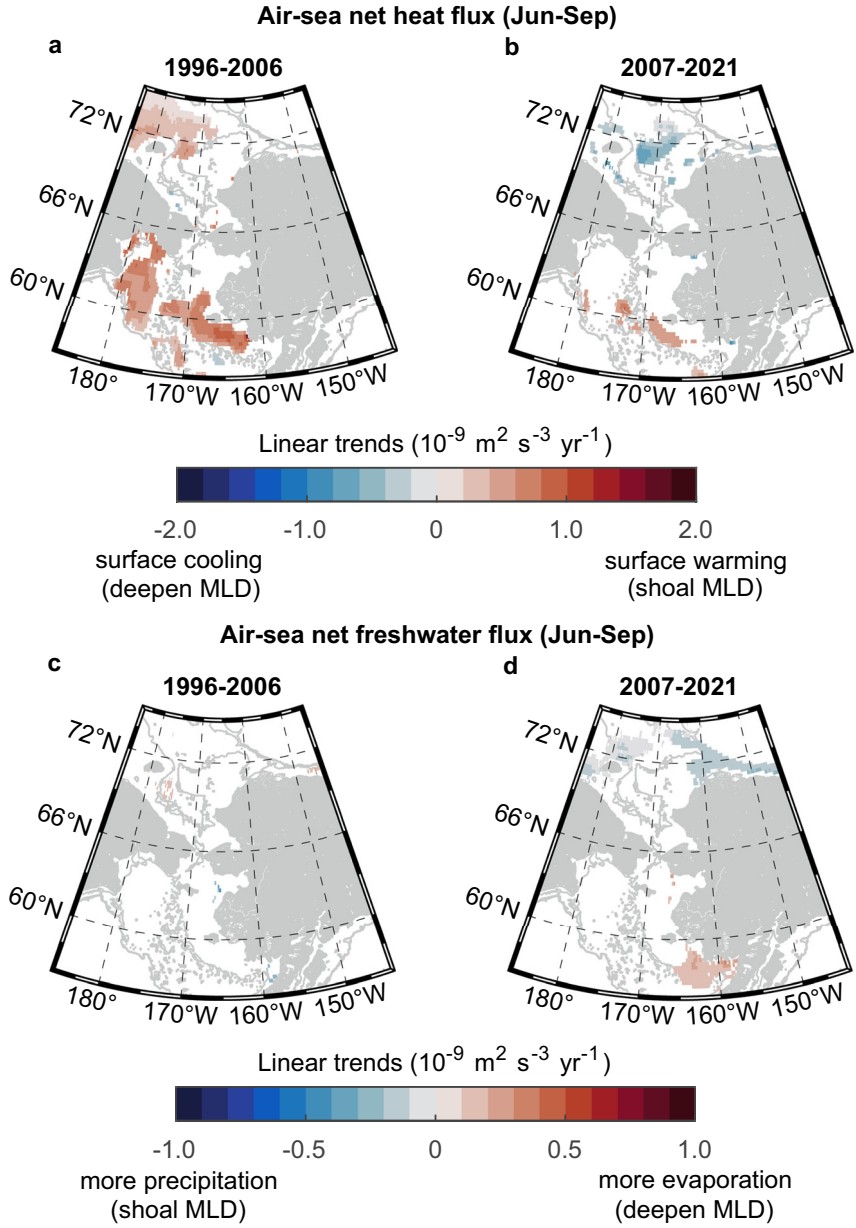

**Fig. 3 | Linear trends of air–sea buoyancy fluxes associated with heat and freshwater.** Linear trends of net air–sea heat flux (positive downwards) from June to September in each 0.25° × 0.25° grid cell based on ECMWF Reanalysis v5 (ERA5) during **a** 1996–2006 and **b** 2007–2021. Linear trends of net air–sea freshwater flux (evaporation minus precipitation) from June to September during **c** 1996–2006 and **d** 2007–2021. Only significant trends with $p < 0.1$ are shown in blue (negative values) to red (positive values) colors. Surface warming and precipitation add buoyancy to the upper ocean, while surface cooling and evaporation cause the upper ocean to lose buoyancy (see the "Methods" section). Gray lines represent isobaths of 50, 250, and 500 m. Source data are provided as a Source Data file.

the influence of sea-ice-derived meltwater in summertime significantly weakened during 2007–2021 because of earlier ice retreat (Fig. 4f). In summary, the reverse trends of the mixed layer after 2007 were likely attributed to a combination of enhanced wind mixing, elongated open water duration, and reduced ice meltwater input.

## Discussion

This study demonstrated that the changing upper ocean dynamics in the Pacific Arctic shelf seas are closely related to sea ice regime shifts within the last three decades. Due to increasing sea-ice-derived meltwater, surface warming, and weakening wind, the summer mixed layer showed a shoaling trend in the first stage before 2006 (Fig. 5a and c). As the pattern of thinner and seasonal sea ice became a new normal after 2007, the exposure of open water to surface wind stress was

greatly extended, which resulted in a deepening trend in MLDs facilitated by early ice retreat and enhanced wind stress (Fig. 5b and d). Although long-term observations indicated increasing Pacific water inflow with more heat and freshwater fluxes into the region[67], the dominance of wind mixing in determining the summer mixed layer was evident in 2007–2021.

The traditional viewpoints explained that rapid sea ice loss should enhance upper ocean stratification[27,34] due to upper ocean warming[68] and/or freshening[69,70]. In contrast, as the Pacific Arctic shelves shifted to a new normal with low to almost no summer sea ice after 2007, the upper mixed layer reversed from shoaling to deepening, suggesting that sea-ice-derived meltwater was less effective in maintaining strong summer stratification. Meanwhile, the wind mixing during a prolonged open water period gradually dominated upper ocean dynamics. We

proved that although Arctic sea ice cover continued its downward trend, the regional upper water column may respond differently when wind mixing outcompeted buoyancy forces in summer. Nevertheless, whether the mixed layer deepening in recent years will continue to

develop or is just a transition with sea ice decline remains to be seen. Once the role of buoyancy flux takes a superior position due to excessive heat gain, the stratification might intensify again in a warmer Arctic[71,72]. Alternatively, a recent CMIP6 model ensemble study projected elevated air-sea momentum and energy transfer, increased surface stress, and enhanced vertical mixing in a warmer and less-ice-covered future Arctic[73], which, in principle, corroborates our findings and proposed mechanisms.

The marine ecosystem on the northern Bering and Chukchi shelves is sensitive to the changes in the upper water column processes. The spring bloom timing on the Bering Sea shelf was related to the seasonal ice breakup[74]. The spring blooms could occur earlier with sea ice retreat[75], and marginal ice zone blooms have occurred more frequently in recent years[76]. Strong wind mixing in the open water would deepen the upper mixed layer, bring nutrients into the euphotic zone, and further expand the blooms to the autumn, potentially supporting production over a longer seasonal scale[77]. However, the nitrate deficit in the upper water column may also intensify[13,78] with a changing mixed layer and notable biological utilization[79]. Altered timing and magnitude of phytoplankton bloom[80] and zooplankton grazing[22], as well as nutrient dynamics, bring uncertainties to the foodweb energy flow, which could negatively impact the strength of pelagic-benthic coupling[3,32]. Moreover, sea ice loss and longer open water duration might promote rapid uptake of atmospheric $CO_2$ and amplify seawater acidification[81,82], despite a strong biological uptake in the Chukchi Sea. In summary, this deepened upper mixed layer modulated by large-scale climate change may have long-lasting impacts on biogeochemical cycling, marine organisms, and the ecosystem as a whole in the new normal Pacific Arctic shelf seas.

## Methods

### Ship-based observations

Vertical conductivity–temperature–depth (CTD) and expendable–conductivity–temperature–depth (XCTD) profiles from the NOAA World Ocean Database 2018 https://www.ncei.noaa.gov/products/world-ocean-database supplemented with ship-based observations from multinational Arctic expeditions (Supplementary Tables 1 and 2) are used to analyze the upper water column structure. Only profiles that have an accepted quality-control flag and contain information on position, date, depth, temperature, and salinity are retained. The accuracies of temperature and salinity measurements are 0.001 °C and 0.002 psu, respectively. The profiles with vertical resolution >5 m or without measurements in the upper 10 m are excluded on account of accurately describing the mixed layer (see the "Methods" subsection "Definition of mixed layer depth"). Profiles with noticeable noises or obvious errors of salinity (temperature) are also eliminated. After rigorous quality control procedures, there are a total of 23,320 hydrographic profiles from 1996 to 2021, with higher spatial coverage along the US Alaskan coast (Supplementary Fig. 3).

### Model reanalysis

Global Ocean Ensemble Physics Reanalysis (GLORYS2V4) from the Mercator Ocean is a 3D gridded product for the ocean's physical state estimation. The data were produced by the Nucleus for European Modeling of the Ocean (NEMO3.1), which was constrained with data assimilation of satellite data and in situ observations[83]. GLORYS2V4 is forced by ERA-Interim, the fourth-generation ECMWF atmospheric reanalysis of the global climate, and has been superseded by the ERA5 after the year 2019. GLORYS2V4 provides daily mean potential temperature, salinity, sea ice concentration, and thickness during 1996–2021 with a 1/4° horizontal resolution and 75 vertical levels. The errors include both the instrument and model errors, with a depth uncertainty of 2% and a temperature uncertainty of 0.1–1 °C for different depths. The vertical resolution of the uppermost 40 m water column is less than 5 m. Because the averaged

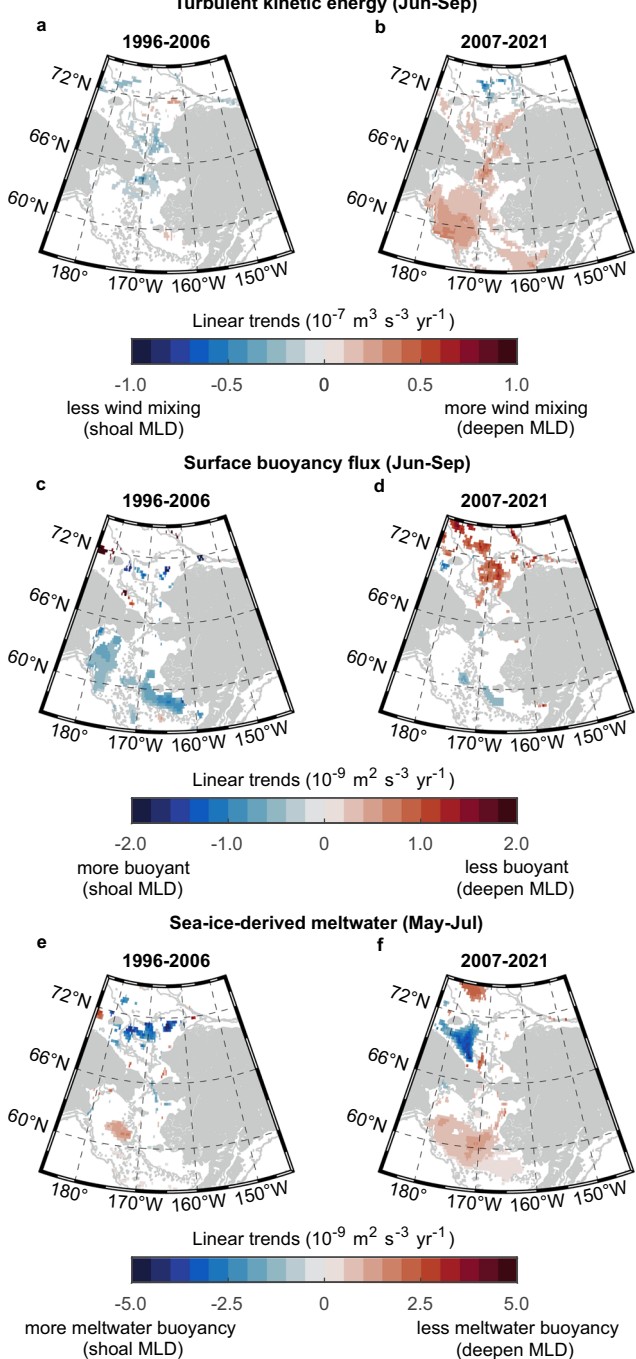

**Fig. 4 | Linear trends of competing factors for changing the upper mixed layer.** Linear trends of wind-induced turbulent kinetic energy from June to September in each 0.25° × 0.25° grid cell during **a** 1996–2006 and **b** 2007–2021. Linear trends of surface buoyancy flux from June to September during **c** 1996–2006 and **d** 2007–2021. Linear trends of ice-meltwater-induced buoyancy flux from June to September during **e** 1996–2006 and **f** 2007–2021. Only significant trends with $p < 0.1$ are shown in blue (negative values) to red (positive values) colors. The wind-induced turbulent kinetic energy and surface buoyancy flux are calculated within the same time window (June–September) as mixed layer depths. The sea-ice-derived meltwater trends are calculated for May–July since the region becomes almost ice-free in August. Gray lines represent isobaths of 50, 250, and 500 m. Source data are provided as a Source Data file.

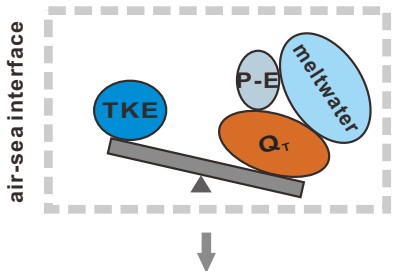

**Fig. 5 | Schematic diagrams showing the dynamic changes of the upper mixed layer under different sea ice regimes. a** Seasonal mixed layer evolution under the climatology of sea ice and **b** the new normal of low sea ice. **c** The battle between wind-induced turbulent kinetic energy (TKE) input and surface buoyancy flux, including air–sea heat ($Q_T$) and freshwater fluxes ($P$–$E$) and sea-ice-derived meltwater in the summertime under the climatology of sea ice and **d** the new normal of low sea ice. The layer from the sea surface to the white dotted line represented the upper mixed layer. The spiral lines below the mixed layer represented the process of turbulent mixing driven by wind stirring. The density of spirals implied the mean strength of turbulent kinetic energy. The variability of monthly wind stress was indicated by different colors, with red representing regional significantly increased, blue representing regional significantly decreased, and gray representing no significant changes. The traditional viewpoints of enhanced upper ocean stratification are caused by surface warming and ocean freshening with rapid sea ice loss (**a** and **c**). Under the new normal of low summer sea ice (**b** and **d**), effective wind mixing in a prolonged open water period triumphed the overall buoyancy forcing with reduced meltwater input, which deepened the mixed layer.

summer MLD based on hydrographic profiles rarely exceeds 35 m, it can be calculated from model results using the same method as observations.

## Sea ice concentration

We calculate the sea ice index using daily sea ice concentration data from 1996 to 2021, which were derived with Nimbus-7 SMMR and DMSP SSM/I-SSMIS with AMSR-E Bootstrap Algorithm[84] and archived by the National Snow and Ice Data Center. This sea ice product has a spatial resolution of 25 km and a general accuracy estimated to be ±3% and ±6% of the actual sea ice concentration in winter and autumn, respectively[84]. The day of the sea ice retreat is defined as the day of a year when the sea ice concentration of a grid cell drops below 15% for three consecutive days. The day of sea ice freeze is defined as the day of a year when the sea ice concentration of a grid cell exceeds 15% for three consecutive days after the annual ice retreat. Open water is defined as a sea ice concentration of <15% and open water duration is calculated as the time between the day of sea ice retreat and freeze. The open water area is the summed area of grid cells with <15% sea ice concentration. The monthly mean sea ice concentration $\overline{\text{SIC}}$ over the study region is calculated based on Eq. (1):

$$\overline{\text{SIC}} = \frac{\sum_{i=1}^{n} \text{sic}_i \times A_i}{\sum_{i=1}^{n} A_i} \quad (1)$$

where $\text{sic}_i$ is the sea ice concentration in grid cell $i$, $A_i$ is the area of grid cell $i$, and $n$ is the total number of grid cells within the study region.

## Atmospheric data

Daily mean atmospheric data are ECMWF reanalysis version 5 (ERA5) with a uniform 0.25° horizontal resolution. The wind stress is calculated from the eastward and northward components of the 10 m wind. The net air-sea heat flux is the sum of shortwave solar radiation, net longwave radiation, latent heat flux, and sensible heat flux (Supplementary Fig. 8a). The rates of evaporation and precipitation are used to compute air-sea net freshwater flux.

## Definition of mixed layer depth

The vertical structural properties of the upper mixed layer are almost uniform, so theoretically, density, salinity, or temperature profiles can all be used as estimators of MLD[85–87]. Among them, density and temperature profiles are more commonly used because salinity profiles tend to be noisier and sparser[88]. The Pacific Arctic Ocean is a typical *beta* ocean, where the water column structure is mainly influenced by salinity[89], so methods based on density rather than temperature are preferable for detecting mixed layer depth. The threshold difference method can provide more accurate MLD estimation than density gradient threshold, least squares regression, and integral methods[87]. Given the simplicity and successful applications[26,27,54,90], this study chooses the density threshold method to estimate MLD. The density is

converted from temperature, salinity, and pressure using the GSW Oceanographic Toolbox (www.TEOS-10.org). The density thresholds of 0.1, 0.125, and 0.15 kg m$^{-3}$ are tested, and the changing tendency of MLD is insensitive to the criterion (Supplementary Fig. 7). Since we focus on the averaged changes of MLD over periods of months and longer, the threshold value of 0.125 kg m$^{-3}$ is deemed appropriate[54,90]. We set the uppermost reference depth of 5 m because CTD profile data are generally less accurate within 0–5 m water depths. The mixed layer depth ($H$) can be inferred from Eq. (2), and $\sigma_0$ was the potential density at the depth referenced to the surface.

$$\sigma_0(z = -H) - \sigma_0(z = -5\,\text{m}) = 0.125\,\text{kg m}^{-3} \quad (2)$$

### 1D sea ice-ocean mixed layer model

The parametric equations of 1D sea ice-ocean mixed layer model[64] simulated the state of the mixed layer with given atmospheric forcing. Based on Lemke's model[64], we input the state of the mixed layer and sea ice from GLORYS2V4 model results to determine the contributions of turbulent kinetic energy and surface buoyancy flux for changing upper ocean (Supplementary Fig. 8b).

The determination of surface buoyancy flux in open sea and under sea ice cover is different. In the open sea, the surface heat flux $Q_T$ can be calculated straightforwardly by the net solar radiation $Q_{sw}$, net longwave radiation $Q_{lw}$, latent heat flux $Q_{lat}$ and sensible heat flux $Q_{sen}$ as Eq. (3).

$$Q_T = \frac{Q_{sw} + Q_{lw} + Q_{lat} + Q_{sen}}{\rho_{water} c_p} \quad (3)$$

where $\rho_{water}$ is seawater density from GLORYS2V4 and $c_p$ is seawater heat capacity. The components of surface heat vertical fluxes are positive downwards (W m$^{-2}$) from ERA5 atmospheric data. The surface freshwater flux $Q_S$ is equal to net air–sea freshwater flux in the open sea, which is determined by the excess of evaporation over precipitation as Eq. (4)

$$Q_S = (E - P) \times \text{MLS} \quad (4)$$

where $E$ and $P$ are the rates of evaporation and precipitation in m s$^{-1}$ from ERA5, MLS is the mixed layer salinity calculated from GLORYS2V4.

The influence of sea ice meltwater was added to the surface freshwater flux $Q_S$ as Eq. (5) for ice-covered regions.

$$Q_S = (E - P) \times \text{MLS} + (\text{MLS} - S_{ice}) \times \text{SIC} \times \frac{dh_{ice}}{dt} \rho_{ice}/\rho_{water} \quad (5)$$

The first term on the right-hand side of Eq. (5) is calculated from ERA5 atmospheric data for the open sea. The second term on the right-hand equation represents the impact of sea ice melting water. The sea ice concentration SIC and melting rate of sea ice thickness $\frac{dh_{ice}}{dt}$ are derived from the model results of GLORYS2V4. Sea ice salinity $S_{ice} = 5$ psu and density $\rho_{ice} = 917$ kg m$^{-3}$ are constants and $\rho_{water}$ is seawater density in the mixed layer.

In the ice-covered regions, the surface heat flux cannot be calculated from solar radiation due to the obstruction of sea ice. According to Lemke's model, the heat buoyancy flux at the sea ice–ocean interface $Q_T$ is given by the entrainment heat flux $B_T$, which is partly determined through the entrainment velocity $W_e$ below the mixed layer. For the 1D model, the entrainment velocity $W_e$ is given by the deepening rate of mixed layer $\frac{dh}{dt}$ and the Ekman pumping/suction

velocity $W_{ek}$ [91,92] as Eq. (6)

$$W_e = \frac{dh}{dt} - W_{ek} \quad (6)$$

Eq. (6) only works for the deepening mixed layer ($\frac{dh}{dt} > 0$). For the shoaling mixing layer, the entrainment velocity $W_e$ is equal to zero. The Ekman pumping/suction velocity $W_{ek}$ is determined by ERA5 10 m wind stress fields and sea ice concentration SIC from GLORYS2V4 model results. Climate Data Toolbox for MATLAB provides the code to estimate the Ekman pumping/suction velocity in ice-covered region.

The change rates of MLT and MLS are dominated by the surface and entrainment fluxes in the mixed layer model. We set an entrainment zone thickness $\delta = 8$ m [93] below the mixed layer, the mean temperature and salinity in the mixed layer (MLT, MLS) and entrainment zone ($T^*$, $S^*$) can be determined by GLORYS2V4 model results. The heat $B_T$ and salt $B_S$ entrainment fluxes can be parameterized with the seawater state in the mixed layer and entrainment zone and entrainment velocity $W_e$ as Eq. (7a) and (7b)

$$B_T = (T^* - \text{MLT})W_e \quad (7a)$$

$$B_S = (S^* - \text{MLS})W_e \quad (7b)$$

MLT is assumed to remain at the freezing point when sea ice is present, so the surface heat flux $Q_T$ is calculated as follows:

$$Q_T = -B_T = -(T^* - \text{MLT})W_e \quad (8)$$

According to the KTN model[62,63], the turbulent kinetic energy TKE provided by wind stirring is balanced with the increase of the potential energy due to the surface fluxes $Q$ and entrainment fluxes $B$, respectively. The surface buoyancy fluxes $Q$ and entrainment buoyancy fluxes $B$ are determined from Eq. (9a) and (9b) with thermal expansion $\alpha$ and haline contraction $\beta$ coefficients.

$$Q = (\beta Q_S - \alpha Q_T)g \quad (9a)$$

$$B = (\beta B_S - \alpha B_T)g \quad (9b)$$

where $g$ is the gravitational acceleration. Assuming a balance between energy production and dissipation integrated over the whole mixed layer in the KTN model, the turbulent kinetic energy induced by wind can be determined from Eq. (10) using an empirical parameter $n = 0.2$[94,95]

$$\text{TKE} = \left(B - \frac{[(1+n)Q - (1-n)|Q|]}{2}\right)h \quad (10)$$

### Statistical analysis of linear trends

In this work, we are interested in detecting long-term trends of the upper mixed layer properties from the ship-based observations and model results. In light of the sparse field data coverage in time and location, we bin MLDs derived from the observations of June–September in each 1° × 2° grid cell to represent the state of the summer upper water column[27,96]. We only use the grid cell with sufficient sampling years to obtain robust decadal trends. For example, the grid cells on the shelf containing fewer than 5 years of data in 1996-2006 were excluded from the linear trend analysis. For 2007–2021, each grid cell required at least 6 years of data. According to the statistical results of Peralta-Ferriz and Woodgate (2015)[27], the summertime (June–September) mixed layer depths in the Chukchi Sea shelf

have a narrow, quasi-normal distribution. Therefore, the long-term trend of MLDs from ship-based observations was estimated as the best linear fit to the summer mean MLDs (June–September). Note that a sudden storm or water mass advection in a short timescale may affect the CTD profiling results on a daily scale, which causes uncertainty to the analyses.

For daily MLDs derived from the GLORYS2V4, we also use the mean values from June to September to calculate the linear trends in each 0.25° × 0.25° grid cell. The linear trends of MLDs in Fig. 2c and d are estimated after removing the seasonal climatology. To get a better comparison with field observations, the linear trends of GLORYS2V4 model results in specific grids (Supplementary Figs. 6 and 7) were calculated without removing the seasonal cycle.

The daily wind stress and air–sea buoyancy flux is derived from ERA5 datasets. The linear trends of wind stress and air–sea net heat and freshwater fluxes from June to September in each 0.25° × 0.25° grid cell are calculated after removing the seasonal climatology. The linear trends of wind-induced turbulent kinetic energy (TKE) are calculated for the time window from June to September. The linear trends of sea-ice-derived meltwater buoyancy are calculated under the time window from May to July since the sea ice started to retreat in May after 2007, and the study region was in open sea in August. The trends of sea ice retreat day and open water duration are calculated for the same 0.25° × 0.25° grid cell.

A linear regression model is used to estimate the trends of variables in each grid cell of the study region, and only significant trends with $p < 0.1$ are shown.

## Data availability
The oceanographic cruise data are publicly available via the Arctic Data Center https://arcticdata.io/catalog/data and Earth Observing Laboratory https://data.eol.ucar.edu/dataset including (Supplementary Table 1) the Arctic Ecosystem Integrated Survey (Arctic Eis), Arctic Observation Network (AON), Bowhead Whale Feeding Ecology Study (BOWFEST), Canada's Three Oceans (C3O), Chukchi Sea Environmental Studies Program (CSESP), Chukchi Sea Offshore Monitoring in Drilling Area (COMIDA), Distributed Biological Observatory (DBO), NASA Impacts of Climate on the Eco-Systems and Chemistry of the Arctic Pacific Environment (ICESCAPE), and Western Arctic Shelf Basin Interactions project (SBI). The CTD casts conducted during the Bering Arctic Subarctic Integrated Survey (BASIS) can be obtained from the NOAA data repository https://oceanview.pfeg.noaa.gov/erddap/search/index.html?page=1&itemsPerPage=1000&searchFor=BASIS. The International Pacific Halibut Commission (IPHC) water column Profiler Data are archived at https://iphc.int/datatest/data/water-column-profiler-data. Data sources from Ecosystems and Fisheries-Oceanography Coordinated Investigations (EcoFOCI) under the Alaska Fisheries Science Center (AFSC) and the Pacific Marine Environmental Laboratory (PMEL) can be retrieved at https://www.ecofoci.noaa.gov/data-links. The oceanographic cruise data from the project Bering Strait: Pacific Gateway to the Arctic are available at the official website http://psc.apl.washington.edu/HLD/Bstrait/Data/BeringStraitCruiseDataArchive.html. Russian–American Long-term Census of the Arctic (RUSALCA) data can be accessed from https://www.ncei.noaa.gov. The Chinese National Arctic Research Expeditions datasets are provided by the National Arctic and Antarctic Data Center https://www.chinare.org.cn. Japan's R/V Mirai cruise data are archived by the Japan Agency for Marine-Earth Science and Technology (JAMSTEC) and can be applied from http://www.godac.jamstec.go.jp/darwin/e/. The Korean cruise data of "Araon" can be obtained from the Korea Polar Data Center https://kpdc.kopri.re.kr/. The Arctic expeditions in Supplementary Table 2 are available at NOAA World Ocean Database 2018 https://www.ncei.noaa.gov/products/world-ocean-database. The ocean model reanalysis of Global Ocean Ensemble Physics Reanalysis (GLORYS2V4) is provided by E.U. Copernicus

Marine Service Information https://doi.org/10.48670/moi-00024. Daily ECMWF reanalysis verion5 (ERA5) atmospheric data are retrieved from the Copernicus Climate Change Service (C3S) Climate Data Store https://cds.climate.copernicus.eu/ (doi:10.24381/cds.adbb2d47). Sea ice concentration data are accessible through https://nsidc.org/data/nsidc-0079/versions/2. Global self-consistent hierarchical high-resolution geography (GSHHG) data https://www.ngdc.noaa.gov/mgg/shorelines/data/gshhs/ and a corrected higher-resolution (2 min) database ETOPO2 https://www.ncei.noaa.gov/products/etopo-global-relief-model are used to draw the coastline and bathymetry of map images with Matlab mapping toolbox. The ship-based observation, GLORYS2V4 model results, and 1D sea ice-ocean mixed layer model results generated in this study have been deposited in the ZENDO database under accession code https://doi.org/10.5281/zenodo.12174955. Source data are provided with this paper.

## Code availability
The MATLAB 2023a is used for data analysis and plotting. All basemaps in the study are generated using the M_Map toolbox (https://www.eoas.ubc.ca/-rich/map.html). Climate Data Toolbox provides the code to estimate the Ekman pumping/suction velocity in the ice-covered region (https://www.mathworks.com/matlabcentral/fileexchange/70338-climate-data-toolbox-for-matlab). Seawater density is converted from temperature, salinity, and pressure using the GSW Oceanographic Toolbox (https://www.teos-10.org/). The MATLAB scripts used for data analysis and 1D sea ice-ocean mixed layer model are available via the ZENODO https://doi.org/10.5281/zenodo.12174955.

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

## Acknowledgements

This work was supported by the National Key Research and Development Program of China (2019YFE0120900 to Z.F., H.J., and Y.W.), National Natural Science Foundation of China (42176225 to Z.F., 42306251 and 42476262 to P.L., 41941013 to J.C., and 42176230 to D.Q.), Shanghai Pujiang Program (20PJ1403100 to Z.F. and 22PJ1406400 to P.L.), Natural Science Foundation of Shanghai (20ZR1416300 to Z.F. and 24ZR1420100 to J.Z.), National Key Research and Development Program of China (2022YFC2808304 to J.Z.), Innovation Program of Shanghai Municipal Education Commission (2021-01-07-00-08-E00102 to H.W. and Z.F.), US National Science Foundation Office of Polar Programs (OPP1917469 to J.M.G.), the National Oceanic and Atmospheric Administration Arctic Research Program of the United States (CINAR 25984.02

to J.M.G.), International Joint Laboratory of Estuarine and Coastal Research, Shanghai (21230750600 to Z.F. and Y.W.), Fujian Provincial Science and Technology Plan (2022J06026 to D.Q.), Shanghai Frontiers Science Center of Polar Science and the Fundamental Research Funds for the Central Universities (to P.L). We are grateful to all the scientists and crew members of numerous Arctic expeditions whose fieldwork and subsequent analysis have significantly contributed to our understanding of the ocean dynamics of ecosystems in the Pacific Arctic Region. We thank Hajime Kawakami, Rebecca Woodgate, Seth Danielson, and Xiaohui Jiao for their assistance in compiling the CTD data.

## Author contributions

Y.W. and Z.F. designed the study, performed analyses, and wrote the manuscript. P.L., H.W., and J.Z. helped optimize the methodology and interpret the physical mechanisms. H.S. contributed to sea ice analysis. H.J., D.Q., and J.M.G. assisted with analyzing the impacts of changing mixed layers on the ecosystem. Z.F., H.J., and J.C. acquired research grants and managed the projects. All authors contributed to improving the manuscript.

## Competing interests

The authors declare no competing interests.
