## [Transparent Peer Review file · Nature Communications]

Enhanced wind mixing and deepened mixed layer in the Pacific Arctic shelf seas with low summer sea ice

Corresponding Author: Professor Zhixuan Feng

Version 0:

Reviewer comments:

Reviewer #1

(Remarks to the Author)

Review of „Enhanced wind mixing and deepened upper mixed layer under the new normal Arctic of low summertime sea ice“

This paper describes the decadal changes in the ocean surface mixed layer depth in the Pacific Arctic. Both observations and reanalysis data are used to quantify the changes and explain the reason. In particular, the authors suggest that the mixed layer in the Pacific Arctic has been deepening since 2007 mainly due to enhanced wind-induced mixing.

Rainville et al. 2011 is one of the first papers that proposed that the Arctic shelf seas could experience deeper mixed layer in summer due to enhanced wind-induced mixing. As the current paper aligns with the idea of those studies, a review of related old studies is needed in the introduction.

One major concern is that the observations in the period of 2007-2021 demonstrate deepening in mixed layer only in 6 grid cells (Fig. 2b), while all other grid cells (data grid cell locations as indicated in Fig. 1c) did not experience statistically significant trends (Fig. 2b). Fig. S3 further shows that all the 4 benthic biological hotspot subdomains did not experience statistically significant deepening trends according to observations in the 2007-2021 period. Instead, only the model reanalysis data show statistically significant trend in three of the four subdomains for this period. Similarly inconsistent with the major claim of the study, in the 1996-2006 period, actually only one of the four subdomains shows shoaling trend in observations and only two of the four subdomains show shoaling trends in the reanalysis (Fig. S3). Considering all these facts, the combination of observations and model reanalysis as presented in the paper does not consistently lead to a convincing conclusion that a regime shift in the upper ocean dynamics in the Pacific Arctic has occurred.

Furthermore, changes in mixed layer depth can be different in different regions of the Arctic Ocean because wind mixing is only one of the influencing factors and dominating factors can differ in different regions and warming scenarios. This point should be made clear and the paper title should reflect that this study is only about Pacific Arctic.

A few other major comments:

Lines 116-120: Are the positive Arctic Oscillation and the Arctic atmospheric dipole relevant to the Pacific Arctic sea ice change? Here the text is intended to explain the shift in sea ice in Pacific Arctic in 2007. In the current text, it is not obvious that the referred atmospheric circulation can explain the sea ice regime shift in the “Pacific Arctic area”.

L132-145: It is not clear why the observed mixed layer depth is highly correlated with the time difference. Instantaneous mixed layer depth can be strongly influenced by storms that had just passed the region. Does the high correlation shown in Fig. 1c indicate that wind-driven mixing is not important for mixed layer depth? Furthermore, why is the negative correlation in the southeastern Bering Sea just due to “alpha” ocean, not due to the impact of wind-induced mixing?

Fig. S2f shows that there is a slight increase in upper ocean salinity in 1996-2006 in the reanalysis. And there is no trend in wind stress in this period. Then how to explain the shoaling trend of mixed layer in the Bering Sea in the reanalysis in this period?

L200-202 These sentences are not clear.

L250-256. Which factor plays the dominant role depends on the exact region. Even if wind-driven mixing plays a major role in Pacific Arctic, it does not prove that the same is true for the Arctic basin area.

In addition, as shown in Fig. S3, the variability among individual observations is a few tens of meters, while the total trend is about 5 meters over the last 15 years in those subdomains with strongest trend in the model (no significant trend in observations though). It would be necessary to discuss the importance of this simulated trend in comparison with the observed strong variability for marine ecosystems.

Ref:

Rainville, L., C.M. Lee, and R.A. Woodgate. 2011. Impact of wind-driven mixing in the Arctic Ocean. *Oceanography* 24(3):136–145, <https://doi.org/10.5670/oceanog.2011.65>.

Reviewer #2

(Remarks to the Author)

Review of 'Enhanced wind mixing and deepened upper mixed layer under the new normal Arctic of low summertime sea ice' by Wang et al.

I find the primary conclusions of this paper to be about the balance between buoyancy forcing and wind stress for a melting Arctic. The conclusions and presentation of the study is overly focused on one time period where wind forcing overcomes buoyancy (2007-2021). The earlier time period (1996-2006) where wind stress did not overcome buoyancy forcing is an equally important result that should be integrated more into the paper. Overall, this is a study worth publishing if improvements are made to better support and present the conclusions.

Major comments:

1) Abstract and title

- a. Neither the abstract nor the title clarify that this paper is focused on the shelf regions exclusively. Reference to 'Pacific Arctic' at line 35 should be more specific; it could mean the entire Western Arctic. Reference to 'biologically productive continental shelves' at line 33 is not sufficient.
- b. Line 39-40 are the main dynamical conclusions of the paper but are incomplete as stated. There is a buoyancy forcing from sea ice melt, and the enhanced wind stress it able to overcome it.
- c. 1996-2006 is completely absent from the abstract, and no reference is given to a shoaling trend over this period or why it occurs.

2) Some key information is missing, which effects how strong the conclusions are and the readability of the paper.

- a. There is not a single example profile of stratification (or nutrients) from this region. It should be included in some form in the main paper in my opinion. Could Figure 4 be based on composite profiles? Is it already?
- b. Please add a timeseries of mixed layer depth to this paper (e.g., similar to Figure 1b for the sea ice). Figure 2 hints at the magnitudes of the mixed layer depth changes, but I'm still left wondering what the actual depths and changes are, especially with respect to the shallow topography. Line 291: 'this deepened upper mixed layer' ... by how much?
- c. Line 148-151: When does the summer mixed layer form? May? June? July? Do all June profiles have a summer mixed layer present? Are only profiles that have a summer mixed layer included in the statistics? Line 248: Explicitly show the time window in this paper. When does the summer mixed layer (potentially) form, and how long is the season over which it develops?

3) The conclusions could be strengthened by making sure the hypothesis and explanations are applicable to both time periods, 1996-2006 and 2007-2021.

- a. Same comment as (1) above, 1996-2006 is completely absent from the abstract.
- b. Line 85-88: As stated, this hypothesis does not explain why the mixed layer shoaled prior to 2007 (when the ice extent was also decreasing).
- c. Line 243-245: The conclusions regarding 1996-2006 need to be clearer. State that buoyancy forcing was sufficient to overcome and wind stress. During this time period, I would say that the 'traditional' view (as stated in this paper) of more melting leads to more stratification / shallower mixed layers is correct.

4) Is the 1996-2006 time period appropriate? How much data is available from 1996-1998? Do the conclusions change if the time period is altered?

- a. Line 128-130 is misleading and suggests that only the data in Table S1 is used. Lines 296-298 contradict this, as data from the world ocean database is also included in addition to the Table S1 data.
- b. A summary table or figure describing the entire dataset is needed. I suggest splitting Figure S5a into two maps, one for each time period, along with splitting Figure S5b into two panels, one for the Chukchi Sea and one for the Bering Sea. A bit more description at line 128-129 would be useful (e.g., total number of summertime profiles within each region and time period).

5) Can the use of wind stress instead of surface stress be justified a bit more? This paper should be addressing the surface stress that the ocean feels accounting for the effects of sea ice cover, and not the wind stress that is imposed on any sea ice present. It's possible the use of wind stress is justified, but that is not clear based on what is presented (analysis of the April

– November time period).

- a. Line 192-194: It seems more appropriate to focus the wind stress calculation on June – September as with mixed layer depths. Why are different months chosen?
- b. Line 214-215 & lines 222-224: Momentum input to the ocean is larger during periods of mixed ice concentration vs. open water. Treating everything as open water isn't a great approach unless it can be justified. What is the ice concentration / coverage like when the summer mixed layer forms?

Additional comments:

- 6) There's a lot of emphasis on what the 'traditional' view is (lines 83, 250), and how this paper contradicts it. I think this needs to be toned down, and simply presented as both buoyancy forcing and wind forcing are potentially important. It will not take away from the significance of this paper.
 - a. I would argue that the 'traditional' view that interannual ice loss can increase buoyancy and shoal the summer mixed layer is in fact correct during 1996-2007.
 - b. Line 37: Maybe rephrase 'contradicting a common notion'. It's an idea, but I don't know that it's pervasive.
 - c. Line 41, 108, and elsewhere: Is this really a regime shift? The change in August ice extent between the two time periods does not qualify in my opinion. Even during 1996-2006, most of the region under consideration was seasonal ice. This contradicts lines 111-113 that claim MYI in the earlier period. I'm a bit confused about this argument.

I do not find Figure 1c to be a new result (lines 136-139). Conceptually, it is true Arctic wide that the summer mixed layer forms in relation to ice melt / melt pond drainage and then gradually deepens during summer. It is not surprising that deeper mixed layers are associated with a longer time since mixed layer shoaling, or that mixed layers are shoaling earlier in the Arctic. The description here seems to have taken a simple concept and made it complicated.

Line 123: define what you mean by trigger time, even if it is stated in the methods. Do lines 123-125 refer to a figure?

Line 133: how is the day of sea ice retreat defined? State briefly even if it stated in the methods.

Line 203-205: What is advection like in this area? The weakening wind stress in the western Bering Sea could induce a change in water properties that is then advected into the region where mixed layer shoaling is observed during 1996-2006. Is this plausible?

Line 304: 'station depths shallower than 10 m' are excluded? This reads like profiles with a measurement in the upper 10 m are excluded, instead of excluding those without a measurement in the upper 10 m.

In table S1, 'Sea storm 2011' is listed as August – Sept 2009 (should be 2011?).

Figure caption labels (a,b,c, etc) should be placed at the start of the sentence describing each panel, not at the end. These are difficult to read.

Version 1:

Reviewer comments:

Reviewer #1

(Remarks to the Author)

This is my second review of this manuscript. The manuscript was substantially improved compared to the original version, especially for the explanation of the MLD changes. However, some places need clarification, and a careful proofreading is still required.

1. Line 37-38: Fig 4e does not show imprint of increased melting in accordance with "massive sea ice loss"
2. Line 40: the MLD increase in the recent period was also contributed by the reduction in sea ice meltwater as shown in Fig 4f. Why is this effect neglected in your conclusion?
3. Line 85-86: if you write "another" here, it would be better to write explicitly at this place what is the first mechanism.
4. L88, role ... for
5. L98: do you mean that sea ice became seasonal during this period?
6. L99, most area of the ...
7. L102, what is "tipping point" in this context?
8. L105, which regions do you refer to here, for "enhanced anomalous oceanic heat flux and amplified ice–albedo feedback"
9. L107, is it "variability" that caused long-term trend?
10. L109, -> day of a year
11. L119, I do not see what is excluded in (a)
12. 126,144, do you mean "the day when ice is free"?
13. 129, With no - without (and at other places)
14. 128, grid - grid cell (and at other places)
15. L137, what does "it" refer to?
16. 145, remove "a"
17. 147, into ... - with sea ice retreat

18. 153, mixed layer - MLD
 19. 166, and then ... - and then the water in the region ...
 20. 170, why "freshening" trend? I would rather say there is no significant salinity trend
 21. 186 models - model results
 22. 208 remove two
 23. 209, - in the season of June ...
 24. 211, why is there growth window for wind stress?
 25. 212, - a positive trend
 26. 213-214: do you mean that "winds" are strengthened by surface warming? The strengthening in winds is different from a strengthening in wind stress.
 27. 215-216: The Arctic Dipole has two active centers, not just "anticyclonic"
 28. 217: You did not illustrate "wind growth" or increase of wind speed in the paper. Wind stress is not only influenced by winds and an increase in wind stress does not prove an increase in wind speed.
 29. Fig S9 does not show a systematic increase in wind stress in June to Sept in the latter period, while the TKE increased more substantially. Why?
 30. 227, - their mixtures?
 31. 229, - other water ...
 32. L230, could you explain what variables are 3-month-mean in your analysis? In Method, you mentioned that daily data were used.
 33. L236-237, this rather tells that changes in ice meltwater play a role! Or, without the changes in sea ice meltwater, MLD in the latter period would have been different. So it is not just that wind mixing played the role.
 34. 241-242, - Linear trends of buoyancy flux associated with ...
 35. 264, for the first period, you did not show "increasing sea-ice-derived meltwater"
 36. 268, - trend in MLD
 37. 267-269, please rewrite this sentence properly
 38. 270, - fluxes into the ...
 39. 274, due to upper ocean warming?
 40. 277, as commented above, the reduction in meltwater could contribute to the increase in MLD
 41. 303, showed – showing
 42. the description of the 1D model should be improved. The most importantly, it should explain the solution procedure, what is done/solved first, the second step, the third
 43. 594, 596, how are $T_{Sbottom}$ and d_T, d_S determined?
 44. 607, the left and right sides of the equation have different units. Time should be divided on the left side.
 45. What is ΔS_{whole} ? How is it determined?
 46. L611, the units on the left and right sides of the equations are different
 47. 625, time should be divided to get "rate"
 48. 630, mixed layer – seawater in the mixed layer
- Please proofread the text thoroughly.

Reviewer #2

(Remarks to the Author)

The manuscript has been improved and I am in favor of publication after two issues are addressed.

Major comments

'Pacific Arctic Ocean' is a much larger region than what is studied here and includes the entirety of the Canada Basin (to ~85°N). I suggest replacing with 'Northern Bering and Chukchi Seas' or 'Pacific Arctic Shelf Seas' throughout. This is a problem in the title and lines 49, 95, 196, 262, 301, and likely other places I have missed.

Lines 159-171: The updated figure 2 is very useful, and shows that most locations do not correspond to a significant trend for mixed layer depth (e.g., 69 grid boxes out of 76 grid boxes). But Figure 2 is presented focusing on only those few boxes with a significant trend, which I think is misleading. Too much of a concrete conclusion is stated from very inconsistent evidence. A more honest description of these results needs to be included. This does not affect the conclusions that can be drawn from the paper overall, as these are primarily based on GLORYS and ERA5 analysis.

Additional comments

Line 35: 'its development' should be more specific, e.g., its interannual to decadal evolution

Line 85: 'another robust mechanism' and 'understanding of the phenomenon' I'm not sure what you're referring to here.

Line 153: suggest changing to 'trends of the mean summer...' so that it's clear how trends are calculated without referring to the methods.

Line 152-161: It would be useful to state here that the mixed layer shoals in May, if this is true for the entire region. I am getting this impression from other parts of the paper, but it would be good to clarify here. This would clarify that even though the mixed layer is shoaling earlier with time, that does not affect the trend statistics during June – September.

Line 182-185: where is the 'inner southeastern Bering shelf'? A box or arrow in Figure 2 would help.

Figure 5a-b: What is indicated by the 'E' and downward arrow, Ekman pumping? For winds and Ekman pumping, it would be better to indicate overall values for the summer period somehow as a monthly analysis (Figure S9) shows no increase during e.g., September. Or indicate the monthly variability depicted in Figure S9 in this summary schematic.

Figure 5c-d: only a suggestion - in panels c and d, it would be more intuitive to keep the terms (TKE, meltwater etc.) on the same side of the balance in both panels, and change which side is up / down between the two.

Line 530-541: GLORYS is used to show the mixed layer depth change, but ERA5 for diagnosis of what's causing that change. Does GLORYS use ERA5 forcing?

Line 648 – 658: Am I correct that for the observations, the mean value from June – September in each year is first calculated, and then a linear trend of those mean values is determined? How does the seasonal cycle bias these results, if observations are in June for one year and September for another? Does this explain why so many grid boxes have an insignificant trend?

Supplement line 39: typo on '2007-201'

Version 2:

Reviewer comments:

Reviewer #1

(Remarks to the Author)

The authors have addressed all my questions and I suggest the paper can be accepted for publication.

(Remarks on code availability)

I had a brief look at the code. The matlab routines have a good readability and are useful for the community.

Reviewer #2

(Remarks to the Author)

I recommend publication of this paper, and have no further comments.

(Remarks on code availability)

I briefly viewed that the linked code existed but will not download and view it further. This appears sufficient.

Response to Reviewers

Reviewer #1 (Remarks to the Author):

Major comments:

1) Rainville et al. 2011 is one of the first papers that proposed that the Arctic shelf seas could experience deeper mixed layer in summer due to enhanced wind-induced mixing. As the current paper aligns with the idea of those studies, a review of related old studies is needed in the introduction.

Response: Thank you for your suggestion. We have updated the introduction and added relevant references as per your suggestion (**References 29, 30, 31; Lines 1129-1134**): **Lines 104-108:** “Previous studies suggested that the mixed layer depth (MLD) in the western Arctic, especially in the Canada Basin, showed a shoaling trend from the 1970s to 2000s²⁶⁻²⁸. At the same time, cruise observations²⁹⁻³⁰ on a short time scale revealed that wind-induced mixing was enhanced and the mixed layer deepened in an increasingly ice-free Arctic Ocean³¹.”

2) One major concern is that the observations in the period of 2007-2021 demonstrate deepening in mixed layer only in 6 grid cells (Fig. 2b), while all other grid cells (data grid cell locations as indicated in Fig. 1c) did not experience statistically significant trends (Fig. 2b). Fig. S3 further shows that all the 4 benthic biological hotspot subdomains did not experience statistically significant deepening trends according to observations in the 2007-2021 period. Instead, only the model reanalysis data show statistically significant trend in three of the four subdomains for this period. Similarly inconsistent with the major claim of the study, in the 1996-2006 period, actually only one of the four subdomains shows shoaling trend in observations and only two of the four subdomains show shoaling trends in the reanalysis (Fig. S3). Considering all these facts, the combination of observations and model reanalysis as presented in the paper does not consistently lead to a convincing conclusion that a regime shift in the upper ocean dynamics in the Pacific Arctic has occurred.

Response: Thank you for pointing out the limitations of the sparse observations and the inconsistency between the model and observation. In the Pacific Arctic region, only limited ship-based field observations were conducted, mostly in the summer months, so CTD/XCTD profile data were sparse. Nevertheless, observed hydrographic profiles are invaluable data to study ocean dynamics. We modified Fig. 1c and removed coastal grids with total water depths of less than 15 m. All grids with colors or crosses had at least 20 hydrographic profiles from 1996 to 2021. Despite that, not all marked grids in Fig. 1c have sufficient data to calculate linear trends of mixed layer depths in 1996-2006 versus 2007-2021. To distinguish geographical locations with insufficient profiles versus no statistical significance, we replotted Fig. 2a and 2b (**Line 568**). The grids with significant linear trends ($p < 0.1$) are shown in colored squares, and the grids with no significant trends are illustrated by black crosses; the blank grids are without sufficient profiles. Based on the new results, the ship-based mixed layer indicates 12 deepening grids in 1996-2006 (Fig 2a) and 7 shoaling grids in 2007-2021 (Fig 2b).

The relevant texts regarding model and observation comparisons have been rewritten (**Lines 433-447**): “To confirm the trends revealed by unevenly distributed hydrographic profiles, we also analyzed daily temperature and salinity output from the Global Ocean Reanalysis and Simulations (GLORYS2V4) and evaluated the mixed layer properties using identical methods (Supplementary Fig. 1c). The reverse MLD trends on the northern Bering and Chukchi shelves during the two periods were broadly consistent with the observations (Fig. 2 and Supplementary Fig. 6a & 7a). The mixed layer shoaled at a mean rate of -0.47 m yr^{-1} ($p < 0.1$) in 1996-2006 and then deepened at a mean rate of 0.33 m yr^{-1} ($p < 0.1$) after 2007. Due to the randomness of ship-based observations and model errors, linear trends of the MLDs based on GLORYS2V4 were more moderate. Lacking tidal mixing processes in the GLORYS2V4 might result in its poor performance on the Bering shelf (Supplementary Fig. 6b & 7b). Observation datasets revealed that the inner southeastern Bering shelf had an overall shoaling trend (Fig. 2b). In comparison, GLORYS2V4 model results showed no obvious trends in the same region (Fig. 2d). For the high-latitude Chukchi Sea, the summer MLD trends derived from observations were similar to the models.”

In the revision, we updated GLORYS2V4 by adding additional model simulation data for 2021, which was made available by the Copernicus database in recent months.

To further demonstrate and compare mixed layer depth calculations from the ship-based observation and model reanalysis data, we added annual mean summer mixed layer depths and linear trends at specific sites (Supplementary Fig. 5 and Fig. 6). The mixed layer depths derived from observations and GLORYS2V4 are generally consistent with each other in the northern Bering and Chukchi Sea shelf during both shoaling (1996-2006) and deepening periods (2007-2021).

3) Furthermore, changes in mixed layer depth can be different in different regions of the Arctic Ocean because wind mixing is only one of the influencing factors and dominating factors can differ in different regions and warming scenarios. This point should be made clear and the paper title should reflect that this study is only about Pacific Arctic.

Response: Thank you for the suggestion. We agree that the title should reflect the exact study region, which has been revised accordingly. The revised title is: “Enhanced wind mixing and deepened upper mixed layer in the new normal Pacific Arctic Ocean with low summertime sea ice” (**Lines 1-2**). We also agree that wind mixing is only one of the influential factors, and dominant factors differ in location and period. The abstract has been revised to emphasize this key point (**Lines 34-38**): “We attribute mixed layer deepening to markedly lengthened open water duration and enhanced wind-induced mixing. Together, these factors increase cumulative wind energy input to an ocean less buffered by sea ice. It is concluded that the relative importance of surface buoyancy flux versus wind forcing switched in two periods despite continued sea ice decline.”

In the revision, we carefully examined the relative contributions of surface buoyancy flux versus wind mixing in summer mixed layer evolution in the subsection “Roles of

surface buoyancy flux versus wind mixing” (Lines 575-685). The diagnostic 1-D mixed layer model has been updated by using a 1-D sea ice-ocean mixed layer model and including sea-ice meltwater as an additional buoyancy source (Lines 1368-1454).

4) Lines 116-120: *Are the positive Arctic Oscillation and the Arctic atmospheric dipole relevant to the Pacific Arctic sea ice change? Here the text is intended to explain the shift in sea ice in Pacific Arctic in 2007. In the current text, it is not obvious that the referred atmospheric circulation can explain the sea ice regime shift in the “Pacific Arctic area”.*

Response: The decadal-scale variability of the atmospheric circulation can explain the sea ice behavior in the Pacific Arctic Ocean (Bi et al., 2023). The Arctic Dipole (AD) index was slightly negative before 2007 and became increasingly positive from 2007 to 2021. The positive AD drives an enhanced anticyclonic Beaufort Gyre and Transpolar Drift and results in increased heat advection into the Arctic (Bi et al., 2021; Polyakov et al., 2023). The AD was also a major driver of the second record-low sea-ice extent in summer 2007 during the satellite record (Wang et al., 2009). The Arctic Oscillation (AO) index has a statistically significant positive trend during the last 60 years (Jeong et al., 2022). However, from the late 1980s to the early 2010s, the AO index showed a significant negative trend (Cohen et al., 2012). This negative trend of AO on the decadal time scale was associated with the Arctic amplification and increasing ice loss (Lindsay et al., 2009; Cohen et al., 2014). Referred atmospheric circulation can explain the shift in the sea ice regime on the Pacific Arctic shelves.

We have rewritten the main text in the subsection “Sea Ice Regime Shift” to better demonstrate the impact of atmospheric circulation on sea ice changes (Lines 162-180): “The sea ice in the Pacific Arctic Ocean often started retreating at the end of March and completely melted or reached the annual minimum extent in September^{6,36,37}. After 2007, this region became almost ice-free in August (Fig. 1a & 1b). The period from 1996 to 2006 could be considered a transition with largely seasonal ice in the Pacific Arctic^{5,38,39}, and a “new normal” of extremely low summertime sea ice generally stabilized after 2007⁴⁰⁻⁴². Most northern Bering and Chukchi continental shelves did not show an early ice retreat or prolonged open water duration from 1996 to 2006 (Supplementary Fig. 2a & 2b). The year 2007 was a potential tipping point, triggered by the transition of a negative Arctic Dipole index to a positive one⁴³. The increasingly positive Arctic Dipole index during 2007-2021 drove an enhanced anomalous oceanic heat flux and amplified ice–albedo feedback^{44,45}. Besides, the negative trend of Arctic Oscillation on the decadal time scale was also associated with increasing ice loss^{46,47}. The atmospheric circulation variability caused the seasonal sea ice to decline further over the long term without any sign of recovery⁴⁸. The trigger time of ice retreat (i.e., year day when sea ice concentration drops below 15% for three consecutive days) advanced by a mean rate of -2.6 days yr⁻¹ ($p < 0.1$) in the Bering Sea and -3.0 days yr⁻¹ ($p < 0.1$) in the Chukchi Sea in 2007-2021 (Supplementary 2c). The open water duration (i.e., periods with sea ice concentration below 15%) increased significantly with a mean rate of +2.8 days yr⁻¹ ($p < 0.1$) in most areas (Supplementary 2d).”

Bi, H. et al. Influences of summertime Arctic Dipole Atmospheric Circulation on sea ice concentration variations in the Pacific sector of the Arctic during different Pacific Decadal Oscillation Phases. *Journal of Climate*, **34**(8): 3003-3019 (2021).

Bi, H., Liang, Y. & Chen, X. Distinct Role of a Spring Atmospheric Circulation Mode in the Arctic Sea Ice Decline in Summer. *Journal of Geophysical Research: Atmospheres*, **128**(6), e2022JD037477 (2023).

Polyakov, I. V. et al. Fluctuating Atlantic inflows modulate Arctic atlantification. *Science* **381**, 972 (2023).

Wang, J. et al. Is the dipole anomaly a major driver to record lows in the Arctic sea ice extent? *Geophysical Research Letters*, **36**(5): L05706 (2009).

Jeong, Y. C., Yeh, S. W., Lim, Y. K., Santoso, A. & Wang, G. Indian Ocean warming as key driver of long-term positive trend of Arctic Oscillation. *npj Clim Atmos Sci.* **5**, 56 (2022).

Cohen, J. et al. Arctic warming, increasing snow cover and widespread boreal winter cooling. *Environ. Res. Lett.* **7**, 014007 (2012).

Cohen, J. et al. Recent Arctic amplification and extreme mid-latitude weather. *Nat. Geosci.* **7**, 627–637 (2014).

Lindsay, R. W., Zhang, J., Schweiger, A., Steele, M. & Stern, H. Arctic sea ice retreat in 2007 follows thinning trend. *Journal of Climate*, **22**(1): 165-176 (2009).

5) L132-145: It is not clear why the observed mixed layer depth is highly correlated with the time difference. Instantaneous mixed layer depth can be strongly influenced by storms that had just passed the region. Does the high correlation shown in Fig. 1c indicate that wind-driven mixing is not important for mixed layer depth? Furthermore, why is the negative correlation in the southeastern Bering Sea just due to “alpha” ocean, not due to the impact of wind-induced mixing?

Response: We agree that storms can quickly deepen mixed layers, as Rainville et al. (2011) cruise observation had indicated. The presence of sea ice largely impedes wind energy into the ocean and can buffer storm impact on the upper ocean. As sea ice melts, wind-induced turbulence can strongly affect the upper ocean structure. Theoretically, a longer open water duration would extend the time window for wind mixing. The positive correlations imply that cumulative wind energy input to the upper ocean becomes even more important than ice meltwater and surface buoyancy flux in determining mixed layer depth, as the 1-D sea ice-ocean mixed layer model diagnosis has revealed (see the subsection “**Roles of surface buoyancy flux versus wind mixing**”). Southeast Bering Sea has a very short time window of sea ice cover, typically only in the winter (see **Fig 1a** blue lines for March ice edge), and therefore, sea ice is less influential on that region. According to Carmack (2007), the upper layer stratified by heat is an alpha ocean (i.e., thermal dominance), and stratified by salinity is a beta ocean (i.e., saline dominance). The southeastern Bering shelf is an alpha ocean where the heat flux input to the ocean is a major factor for stratification (Ladd and Stabeno, 2012; Stabeno et al., 2012).

The paragraph has been rewritten to better explain the correlations of mixed layer depth and the time difference between hydrographic sampling and ice retreat (**Lines 211-285**). “The states and dynamics of the upper water column might have been altered with continuous sea ice decline and longer ice-free days. We compiled 23,320 observed CTD/XCTD profiles from multiple cruises (Supplementary Table 1 & 2) in 1996-2021 to quantify the upper mixed layer in the study region (see Methods for MLD definition; See Supplementary Fig. 1b for regional examples of potential density profiles and MLDs). Although no significant correlation was found between observed MLD and sea ice concentration, it was closely related to the trigger time of sea ice retreat (Fig. 1c). We defined the timing difference between the sampling time of hydrography and sea ice retreat day in the corresponding year as Δt and retained 19,742 profiles with positive Δt values (i.e., hydrographic profiles were sampled after sea ice retreat) to calculate their correlations with MLDs in each grid (see Methods). The timing difference Δt was positively correlated with the summer MLD in most of the Bering and Chukchi continental shelves (Fig. 1c). In other words, the closer hydrographic profiling to annual ice retreat, the shallower mixed layer with more content of sea-ice-derived meltwater. Theoretically, the mixed layer would deepen under a sufficiently long wind stirring after sea ice retreat, and ice meltwater became less and less influential into the ice-free period. Notably, a negative correlation was found in the sub-arctic southeastern Bering shelf, indicating that the physical mechanism that controls upper ocean dynamics there could be different from other regions of the Pacific Arctic.”

6) *Fig. S2f shows that there is a slight increase in upper ocean salinity in 1996-2006 in the reanalysis. And there is no trend in wind stress in this period. Then how to explain the shoaling trend of mixed layer in the Bering Sea in the reanalysis in this period?*

Response: Both upper ocean salinity and temperature increased in 1996-2006 based on the GLORYS2V4 reanalysis (Supplementary Fig. 4b & 5b). An increase in salinity may deepen the mixed layer, while temperature increases shoal the mixed layer. Although there were no significant trends in wind stress (Supplementary Fig. 9) and wind-induced turbulent kinetic energy (Fig. 4a) on the southern Bering Sea shelf from 1996 to 2006, this region may have received more surface buoyancy fluxes during this period (Fig 4c). Therefore, increasing atmospheric heat flux input was a dominant factor that caused the shoaling trend of mixed layer depth on the southern Bering shelf in 1996-2006.

7) *L200-202 These sentences are not clear.*

Response: The original L200-202 sentence within the whole paragraph was to explain how and why wind patterns changed based on our and previous studies. The Arctic winds are integral to heat and energy transfer between the atmosphere and the ocean (Hughes and Cassano, 2015). The surface roughness of sea ice is usually higher than that of open water, so winds tend to strengthen where sea ice diminishes (Jakobson et al., 2019). Besides, the increased surface temperature and turbulent heat flux will decrease the low-level atmospheric stability as sea ice loss. The decrease in static stability further promotes stronger winds (Mioduszewski et al., 2018). In addition, the large decrease in sea level pressure (Aleutian low) also results in a mean geostrophic

wind speed increase (Wang et al., 2021; Huang et al., 2022). The growth of wind in the transition from sea ice to open ocean is associated with increased surface warming, reduced surface roughness, decreased static stability, and lowered sea level pressure.

This paragraph has been rewritten (**Lines 585-598**):

“Local winds over the Pacific Arctic could account for 30-50% of the synoptic scale variability⁵⁰⁻⁵², and explain 10-20% MLD variance^{27,53}. We evaluated monthly mean wind stress from May to September that spanned ice melting and open water periods based on ERA5. Significant changes of wind stress mostly occurred in open sea or marginal ice zone for both two periods (Supplementary Fig. 9). The wind stress from the northern Bering to Chukchi Sea shelves decreased from June to August during 1996-2006 with a mean rate of $-1.2 \times 10^{-3} \text{ Pa yr}^{-1}$. With earlier sea ice retreat during the later period than the previous, the wind stress had a nearly 4-month growth window starting in May with a positive rate of $1.7 \times 10^{-3} \text{ Pa yr}^{-1}$, which agreed with prior results of climate models^{54,55} and observations⁵⁶. The strengthened wind stress resulted from sea ice loss and surface warming^{54,57,58}, which decreased ocean surface roughness and atmospheric stability. The positive Arctic Dipole atmospheric circulation has been more anticyclonic since 2007^{59,60}, which increased the pressure gradient between the Aleutian Low and Beaufort High⁶⁰ and further promoted wind growth⁶¹.”

Hughes, M., & Cassano, J. J. The climatological distribution of extreme Arctic winds and implications for ocean and sea ice processes. *J. Geophys. Res. Atmos.* **120**, 7358–7377 (2015).

Jakobson, L., Vihma, T., & Jakobson, E. Relationships between sea ice concentration and wind speed over the Arctic Ocean during 1979–2015. *Journal of Climate*, **32**, 7783–7796 (2019).

Mioduszewski, J., Vavrus, S., & Wang, M. Diminishing arctic sea ice promotes stronger surface winds. *Journal of Climate*, **31**(19), 8101–8119 (2018).

Wang, Q., Danilov, S., Mu, L., Sidorenko, D., & Wekerle, C. Lasting impact of winds on Arctic sea ice through the ocean’s memory. *The cryosphere*, **15**(10), 4703-4725 (2021).

Huang, Y., Bai, X., & Leng, H. Synoptic-scale variability in the Beaufort High and spring ice opening in the Beaufort Sea. *Frontiers in Marine Science*, **9**, 929209 (2022).

8) L250-256. *Which factor plays the dominant role depends on the exact region. Even if wind-driven mixing plays a major role in Pacific Arctic, it does not prove that the same is true for the Arctic basin area.*

Response: Thanks for the suggestion. We have revised both the title and main text to specify our study region as the Pacific Arctic Ocean. This paragraph has been rewritten to further clarify the point (**Lines 748-987**):

“The traditional viewpoints explained that rapid sea ice loss should enhance upper ocean stratification^{27,34} due to climate warming⁶⁸ and/or ocean freshening^{69,70}. In

contrast, as the Pacific Arctic shelves shifted to a new normal with extremely low summer sea ice after 2007, the upper mixed layer reversed from shoaling to deepening, suggesting that sea-ice-derived meltwater played a secondary role, whereas perfectly-timed wind stirring during prolonged open water period dominated upper ocean dynamics. We proved that although Arctic sea ice cover continued its downward trend, the regional upper water column may respond differently when wind-induced mixing outcompeted buoyancy forces in summer. Nevertheless, whether the mixed layer deepening in recent years will continue to develop or is just a transition with sea ice decline remains to be seen. Once the role of buoyancy flux takes a superior position, the stratification might intensify again in a warmer and ice-free future Arctic^{71,72}.”

9) In addition, as shown in Fig. S3, the variability among individual observations is a few tens of meters, while the total trend is about 5 meters over the last 15 years in those subdomains with strongest trend in the model (no significant trend in observations though). It would be necessary to discuss the importance of this simulated trend in comparison with the observed strong variability for marine ecosystems.

Response: Thanks for your suggestion. We have made the following changes in the revision. The new Supplementary Fig 1 presents the upper ocean structures at four different Distributed Biological Observatory boxes using both observations (Supplementary Fig 1b) and GLORYS2V4 reanalysis (Supplementary Fig 1c). Supplementary Fig. 6 and Fig. 7 illustrate annual mean mixed layer depths at multiple sites in 1996-2006 and 2007-2021. The changes in upper ocean physical processes on the marine ecosystem are rewritten (**Lines 988-1004**):

“The marine ecosystem on the northern Bering and Chukchi shelves is sensitive to the changes in the upper water column processes. The spring bloom timing on the Bering Sea shelf was related to the seasonal ice breakup⁷³. The spring blooms could occur earlier with sea ice retreat⁷⁴, and marginal ice zone blooms have occurred more frequently in recent years⁷⁵. Strong wind mixing in the open water would deepen the upper mixed layer, bring nutrients into the euphotic zone, and further expand the blooms to the autumn, potentially supporting production over a longer seasonal scale⁷⁶. However, the nitrate deficit in the upper water column may also intensify^{13,77} with a changing mixed layer and notable biological utilization⁷⁸. Altered timing and magnitude of phytoplankton bloom⁷⁹ and zooplankton grazing²², as well as nutrient dynamics, bring uncertainties to the foodweb energy flow, which could negatively impact the strength of pelagic-benthic coupling^{3,32}. Moreover, sea ice loss and longer open water duration might promote rapid uptake of atmospheric CO₂ and amplify seawater acidification^{80,81}, despite a strong biological uptake in the Chukchi Sea. In summary, this deepened upper mixed layer modulated by large-scale climate change may have long-lasting impacts on biogeochemical cycling, marine organisms, and the ecosystem as a whole in the new normal Pacific Arctic.”

Reviewer #2 (Remarks to the Author):

Major comments:

1) a. *Neither the abstract nor the title clarify that this paper is focused on the shelf regions exclusively. Reference to 'Pacific Arctic' at line 35 should be more specific; it could mean the entire Western Arctic. Reference to 'biologically productive continental shelves' at line 33 is not sufficient.*

b. *Line 39-40 are the main dynamical conclusions of the paper but are incomplete as stated. There is a buoyancy forcing from sea ice melt, and the enhanced wind stress it able to overcome it.*

c. *1996-2006 is completely absent from the abstract, and no reference is given to a shoaling trend over this period or why it occurs.*

Response: Thank you for the suggestion! We agree that the title and abstract should reflect the exact study region, which has been revised accordingly. The revised title is: "Enhanced wind mixing and deepened upper mixed layer in the new normal Pacific Arctic Ocean with low summertime sea ice" (**Lines 1-2**).

We also agree that the abstract should emphasize changes in the mixed layer depth trends and the relative importance of wind versus surface buoyancy forces over the two periods. The abstract has been rewritten (**Lines 34-51**):

"The Arctic Ocean has undergone dramatic sea ice loss in recent decades, with earlier melting and later freezing. The thickness of multiyear sea ice sharply decreased in the early 2000s, and the ice regime shifted to thinner and more mobile seasonal ice. However, the possible changes in the upper ocean dynamics during this drastic transformation have not been thoroughly investigated, especially for the biologically productive Pacific Arctic shelves. Here, we quantified the summertime upper mixed layer and dynamically related its development with sea ice and atmospheric forcing, using a compilation of hydrographic observations and model reanalysis data from 1996 to 2021. Before 2006, a shoaling summer mixed layer was associated with massive sea ice loss and surface warming. However, the upper mixed layer reversed to a generally deepening trend after 2007. We attribute mixed layer deepening to markedly lengthened open water duration and enhanced wind-induced mixing. Together, these factors increase cumulative wind energy input to an ocean less buffered by sea ice. It is concluded that the relative importance of surface buoyancy flux versus wind forcing switched in two periods despite continued sea ice decline. The changing upper ocean dynamics and dominant forcing factors may profoundly affect the marine ecosystem, which has seen unusual fall blooms and more severe ocean acidification, deeper and northward, in the recent decade."

2) *Some key information is missing, which affects how strong the conclusions are and the readability of the paper.*

a. *There is not a single example profile of stratification (or nutrients) from this region. It should be included in some form in the main paper in my opinion. Could Figure 4 be based on composite profiles? Is it already?*

b. *Please add a timeseries of mixed layer depth to this paper (e.g., similar to Figure 1b*

for the sea ice). Figure 2 hints at the magnitudes of the mixed layer depth changes, but I'm still left wondering what the actual depths and changes are, especially with respect to the shallow topography. Line 291: 'this deepened upper mixed layer' ... by how much? c. Line 148-151: When does the summer mixed layer form? May? June? July? Do all June profiles have a summer mixed layer present? Are only profiles that have a summer mixed layer included in the statistics? Line 248: Explicitly show the time window in this paper. When does the summer mixed layer (potentially) form, and how long is the season over which it develops?

Response: The potential density profiles and corresponding mixed layer depths at multiple sites have been newly added in the supplementary materials as examples (see Supplementary Fig. 1b & 1c). Supplementary Fig. 1b shows the seawater potential density profiles averaged in the summer months (June-September) and other months from 1996-2021 based on ship hydrographic sampling. The original Figure 4 was not based on composite profiles since that schematic was mainly used to illustrate the combined effects of wind, atmospheric buoyancy flux, and ice melting water on the upper mixed layer. Fig. 5 (original Fig 4) has also been redrawn to better emphasize key points of this study.

The time series of mixed layer depths from ship-based observations and GLORYS2V4 reanalysis at specific sites have been added to the supplementary materials (see Supplementary Fig. 6 and Fig. 7). The magnitudes of mixed layer deepening in 2007-2021 can be found in Fig 2b (based on hydrographic observations) and Fig 2d (based on GLORYS2V4 reanalysis), which varied in the range of 0-1.5 m yr⁻¹.

We practically define the summer month as June to September, following the definition of previous studies (Peralta-Ferriz and Woodgate, 2015; Danielson et al., 2020). We did not use a critical depth or state to define the formation of the summer mixed layer but rather focused on the change of the mean summer mixed layer within a fixed annual time window (June-September). All hydrographic profiles sampled between June and September were included in the analysis. The time window “June to September” has been clearly labeled in Fig. 2-Fig. 5.

Peralta-Ferriz, C., & Woodgate, R. A. Seasonal and interannual variability of pan-arctic surface mixed layer properties from 1979 to 2012 from hydrographic data, and the dominance of stratification for multiyear mixed layer depth shoaling. *Progress in Oceanography*, **134**, 19-53 (2015).

Danielson, S. L., et al. Manifestation and consequences of warming and altered heat fluxes over the Bering and Chukchi Sea continental shelves. *Deep-Sea Research II*, **177**, 104781 (2020).

3) The conclusions could be strengthened by making sure the hypothesis and explanations are applicable to both time periods, 1996-2006 and 2007-2021.

a. Same comment as (1) above, 1996-2006 is completely absent from the abstract.

b. Line 85-88: As stated, this hypothesis does not explain why the mixed layer shoaled prior to 2007 (when the ice extent was also decreasing).

c. Line 243-245: The conclusions regarding 1996-2006 need to be clearer. State that buoyancy forcing was sufficient to overcome wind stress. During this time period, I would say that the 'traditional' view (as stated in this paper) of more melting leads to more stratification / shallower mixed layers is correct.

Response: Thank you for this excellent suggestion. We have substantially modified our analyses and texts accordingly. The unified concept and hypothesis on the relative importance of surface buoyancy forces versus wind mixing can explain what happened to the upper mixed layer in both periods.

First, in the Introduction section, we have rewritten the hypothesis and explanations regarding upper mixed layer changes and potential mechanisms (Lines 113-159).

“Here, we analyze 23,320 shipboard CTD/XCTD profiles and model reanalysis results from 1996 to 2021 to depict the interannual variations and decadal trends of the upper ocean properties on the Pacific Arctic shelves. Our results suggest a potential regime shift in the upper ocean dynamics: a reverse trend of MLD from shoaling to deepening after 2007, a well-known year with the second lowest summer sea ice coverage since the satellite monitoring era. This finding demonstrates another robust mechanism and consequence of sea ice loss on the upper ocean dynamics, in addition to a traditional understanding of the phenomenon³³⁻³⁵. We further quantify the roles of wind mixing, atmospheric buoyancy flux, and sea-ice-derived meltwater on summer MLD development and variations. Our hypothesis is that in increasingly ice-free Pacific Arctic shelves with prolonged open water duration, wind forcing can sufficiently mix the upper water column and become a more dominant factor than atmospheric buoyancy and ice meltwater in controlling the summer mixed layer.”

Second, we applied a new 1D sea ice-ocean mixed layer model for in-depth diagnosis. This model incorporated sea-ice-derived meltwater as an additional buoyancy source to the upper ocean. We found that the shoaling mixed layer on the Bering Sea shelf from 1996 to 2006 resulted from increased atmospheric heat flux to the ocean (**Fig 3a**), whereas increased ice meltwater (**Fig. 4e**) strengthened the stratification (mixed layer shoaling) in the northern Chukchi Sea (**Fig. 2c**). These two phenomena and underlying mechanisms were consistent with the “traditional view”.

The abstract has been rewritten to emphasize key mechanisms of shifting forcing factors, wind versus surface buoyancy, associated with sea ice loss (**Lines 42-48**): “Before 2006, a shoaling summer mixed layer was associated with massive sea ice loss and surface warming. However, the upper mixed layer reversed to a generally deepening trend after 2007. We attribute mixed layer deepening to markedly lengthened open water duration and enhanced wind-induced mixing. Together, these factors increase cumulative wind energy input to an ocean less buffered by sea ice. It is concluded that the relative importance of surface buoyancy flux versus wind forcing switched in two periods despite continued sea ice decline.”

4) Is the 1996-2006 time period appropriate? How much data is available from 1996-1998? Do the conclusions change if the time period is altered?

a. Line 128-130 is misleading and suggests that only the data in Table S1 is used. Lines 296-298 contradict this, as data from the world ocean database is also included in addition to the Table S1 data.

b. A summary table or figure describing the entire dataset is needed. I suggest splitting Figure S5a into two maps, one for each time period, along with splitting Figure S5b into two panels, one for the Chukchi Sea and one for the Bering Sea. A bit more description at line 128-129 would be useful (e.g., total number of summertime profiles within each region and time period).

Response: Thanks for pointing out the unclarity and inconsistency of the data description. We have made multiple improvements in the main texts and supplementary information.

First, the numbers of summertime profiles in different regions and periods were stated and plotted. See subsection “Reversal of Mixed Layer Depth Trends” and Supplementary Fig 3. This figure has been modified based on the reviewer’s suggestions (see Supplementary Fig. 3). The detailed description of the dataset has been added to the main text (**Lines 288-293**).

“To assess the impacts of changing sea ice conditions on the upper ocean, we further calculated trends of the summer (June to September) mixed layer using a total of 18,997 quality-controlled hydrographic profiles over two separate periods: 1996-2006 versus 2007-2021 (Supplementary Fig. 3). There were 2,835 summertime profiles in the Bering Sea shelf during 1996-2006 and 6,554 profiles after 2007. The Chukchi Sea shelf contained 2,146 profiles in 1996-2006 and 7,462 in 2007-2021, respectively.”

If we altered the time period to 1996-2006, 1998-2006, and 2000-2006, the shoaling trends of the mixed layer during different periods still existed in the Bering and Chukchi Sea shelf (see Review Fig. 1).

Review Fig. 1 Linear trends of upper mixed layer depth during different periods. The linear trends of mean summer mixed layer depth (June to September) during (a) 1996-2006, (b) 1998-2006 and (c) 2000-2006. Only the linear trends with p-value < 0.1 were shown. The grey line showed isobath of 50m, 250m, and 500m.

The observed CTD/XCTD profiles were compiled from the NOAA World Ocean Database 2018 (see Supplementary Table 2) and multinational Arctic expeditions (see Supplementary Table 1). We have added a summary of ship-based hydrographic profiles from the World Ocean Database to the supplementary information (Supplementary Table 2).

5) Can the use of wind stress instead of surface stress be justified a bit more? This paper should be addressing the surface stress that the ocean feels accounting for the effects of sea ice cover, and not the wind stress that is imposed on any sea ice present. It's possible the use of wind stress is justified, but that is not clear based on what is presented (analysis of the April – November time period).

a. Line 192-194: It seems more appropriate to focus the wind stress calculation on June – September as with mixed layer depths. Why are different months chosen?

b. Line 214-215 & lines 222-224: Momentum input to the ocean is larger during periods of mixed ice concentration vs. open water. Treating everything as open water isn't a great approach unless it can be justified. What is the ice concentration / coverage like when the summer mixed layer forms?

Response: Thank you for asking about the wind stress calculation. We have made multiple adjustments per your suggestions. In the original work, we calculated the mean wind stress within a fixed time window (April to November) to account for both ice-covered and open-water periods. As the reviewer pointed out, this simple calculation may not be appropriate. Those wind stress trend subplots (original Fig 3a & 3b) have been removed. Instead, we have added trend analyses of monthly wind stress (May to September) along with sea ice edge in 1996-2006 versus 2007-2021 (**Supplementary Fig. 9**). The significant changes in wind stress were generally shown in the open sea or marginal ice zone. Also, we did not consider the impact of sea-ice-derived meltwater in the original 1D ocean mixed layer model. In our new analysis, we applied a 1D sea ice-ocean mixed layer model to separate competing factors for the upper mixed layer evolution, including wind mixing (Fig 4a), surface buoyancy flux (Fig 4b), and sea-ice-derived meltwater (Fig 4c). The sea ice condition has been considered in our new diagnosis. All the factors, except ice meltwater, are calculated based on the June to September average, along with mixed layer depths. The sea-ice-derived meltwater trends are calculated for May to July since the whole region becomes almost ice-free in August.

Additional comments:

6) There's a lot of emphasis on what the 'traditional' view is (lines 83, 250), and how this paper contradicts it. I think this needs to be toned down, and simply presented as both buoyancy forcing and wind forcing are potentially important. It will not take away from the significance of this paper.

a. I would argue that the 'traditional' view that interannual ice loss can increase buoyancy and shoal the summer mixed layer is in fact correct during 1996-2007.

b. Line 37: Maybe rephrase 'contradicting a common notion'. It's an idea, but I don't

know that it's pervasive.

c. Line 41, 108, and elsewhere: Is this really a regime shift? The change in August ice extent between the two time periods does not qualify in my opinion. Even during 1996-2006, most of the region under consideration was seasonal ice. This contradicts lines 111-113 that claim MYI in the earlier period. I'm a bit confused about this argument.

Response: We agree with the reviewer's comments. In the revised paper, the statement “contradicting a common notion” has been removed. Instead, we emphasized a key mechanism, which is the relative importance of surface buoyancy flux versus wind forcing, which switched over two periods in the abstract and main text.

The original description of multiyear ice was incorrect. The subsection “Sea Ice Regime Shift” has been rewritten (**Lines 162-169**).

“The sea ice in the Pacific Arctic Ocean often started retreating at the end of March and completely melted or reached the annual minimum extent in September^{6,36,37}. After 2007, this region became almost ice-free in August (Fig. 1a & 1b). The period from 1996 to 2006 could be considered a transition with largely seasonal ice in the Pacific Arctic^{5,38,39}, and a “new normal” of extremely low summertime sea ice generally stabilized after 2007⁴⁰⁻⁴². Most northern Bering and Chukchi continental shelves did not show an early ice retreat or prolonged open water duration from 1996 to 2006 (Supplementary Fig. 2a & 2b).”

7) I do not find Figure 1c to be a new result (lines 136-139). Conceptually, it is true Arctic wide that the summer mixed layer forms in relation to ice melt / melt pond drainage and then gradually deepens during summer. It is not surprising that deeper mixed layers are associated with a longer time since mixed layer shoaling, or that mixed layers are shoaling earlier in the Arctic. The description here seems to have taken a simple concept and made it complicated.

Response: Thank you for the comment. We argue Figure 1c is important for two reasons. First, this simple correlation analysis used pure observation evidence to prove the concept regarding the time evolution of mixed layers after the ice retreat. Second, this figure is useful for logically leading to follow-up analyses of summer mixed layer trends and underlying physical mechanisms.

8) Line 123: define what you mean by trigger time, even if it is stated in the methods. Do lines 123-125 refer to a figure?

Response: The definition of trigger time of ice retreat has been stated in the main content. The original lines 123-125 refer to the results from Supplement Fig. 2c & 2d, marked in the revised paper (**Lines 175-180**).

“The trigger time of ice retreat (i.e., year day when sea ice concentration drops below 15% for three consecutive days) advanced by a mean rate of $-2.6 \text{ days yr}^{-1}$ ($p < 0.1$) in the Bering Sea and $-3.0 \text{ days yr}^{-1}$ ($p < 0.1$) in the Chukchi Sea in 2007-2021

(Supplementary Fig. 2c). The open water duration (i.e., periods with sea ice concentration below 15%) increased significantly with a mean rate of +2.8 days yr⁻¹ ($p < 0.1$) in most areas (Supplementary Fig. 2d).”

9) Line 133: *how is the day of sea ice retreat defined? State briefly even if it stated in the methods.*

Response: See the previous one. The term “day of sea ice retreat” has the same definition as “trigger time of ice retreat”.

10) Line 203-205: *What is advection like in this area? The weakening wind stress in the western Bering Sea could induce a change in water properties that is then advected into the region where mixed layer shoaling is observed during 1996-2006. Is this plausible?*

Response: We agree with the reviewer's comments that the advection of water mass may impact the mixed layer in the region. However, the 1D sea ice-ocean mixed layer cannot account for the advection of water masses. We acknowledged such a limitation (Lines 672-677).

“The Chukchi shelf contained complex water masses, including Alaska Coastal Water, ice meltwater, dense winter water, Bering Sea water, and mixtures in summer⁶⁵. Nevertheless, the timing and extent of the advection of ice meltwater and water masses varied among regions and years⁶⁶. Linear trends of competing factors for changing the upper mixed layer were calculated using three-month-mean variables to smooth water mass advection.”

11) Line 304: *‘station depths shallower than 10 m’ are excluded? This reads like profiles with a measurement in the upper 10 m are excluded, instead of excluding those without a measurement in the upper 10 m.*

Response: The sentence has been modified (Lines 1261-1265).

“The profiles with vertical resolution greater than 5m or without measurements in the upper 10 m are excluded on account of accurately describing the mixed layer (see Methods section ‘Definition of mixed layer depth’).”

8) In table S1, ‘Sea storm 2011’ is listed as August – Sept 2009 (should be 2011?).

Response: The expedition date of ‘Sea storm 2011’ has been corrected in the Supplementary Table 1.

9) Figure caption labels (a, b, c, etc) should be placed at the start of the sentence describing each panel, not at the end. These are difficult to read.

Response: All figure caption labels have been modified per your suggestion.

Response to Reviewers

Reviewer #1 (Remarks to the Author):

1) *Line37-38: Fig 4e does not show imprint of increased melting in accordance with “massive sea ice loss”.*

Response: Thank you for the suggestion. The perennial ice in the Pacific Arctic shelves steadily decreased and reached its second record-low extent in 2007 (Comiso, 2006; Kwok, 2007). Fig 1a and Fig 4e indicated that massive sea ice loss mainly occurred in the northern Chukchi Sea shelf. The sentence has been revised to be more proper (**Lines 35-36**): “Before 2006, a shoaling summer mixed layer was associated with sea ice loss and surface warming.”

Comiso, J. C. Abrupt decline in the Arctic winter sea ice cover. *Geophysical Research letters* **33**, L18504 (2006).

Kwok, R. Near zero replenishment of the Arctic multiyear sea ice cover at the end of 2005 summer. *Geophysical Research letters* **34**, L05501 (2007).

2) *Line 40: the MLD increase in the recent period was also contributed by the reduction in sea ice meltwater as shown in Fig 4f. Why is this effect neglected in your conclusion?*

Response: We agree with the reviewer's comments. In the revised abstract, the role of reduced sea ice meltwater has been emphasized (**Lines 37-39**): “We attribute mixed layer deepening to markedly lengthened open water duration, enhanced wind-induced mixing and ice meltwater reduction.”

3) *Line85-86: if you write “another” here, it would be better to write explicitly at this place what is the first mechanism.*

Response: Thank you for the suggestion. The phenomenon and mechanism of traditional understanding have been clarified in the revise paper (**Lines 83-86**): “This finding demonstrates another robust mechanism and consequence of sea ice loss on the upper ocean dynamics, in addition to a traditional understanding of shoaling mixed layer mainly due to more ice meltwater³³⁻³⁵.”

4) *Line88, role ... for*

Response: The sentence has been modified (**Lines 86-87**): “We further quantify the roles of wind mixing, atmospheric buoyancy flux, and sea-ice-derived meltwater for summer MLD development and variations.”

5) *Line98: do you mean that sea ice became seasonal during this period?*

Response: The annual Arctic sea ice minimum extent declined rapidly during 1996 to 2006. The perennial ice cover in the Chukchi Sea shelf continued to shrink and gradually replaced with seasonal sea ice which grows in the fall and winter, and melts

in the summer. The northern movement of mean summer sea ice edge represents the shrinking of perennial ice cover (see Fig. 1a). The sentence has now been rewritten (**Lines 96-99**): “The period from 1996 to 2006 could be considered as a transition with shrinking perennial ice cover in the Pacific Arctic^{5,38,39}, and a “new normal” of extremely low summertime sea ice generally stabilized after 2007⁴⁰⁻⁴².”

6) *Line99, most area of the ...*

Response: The sentence has been modified (**Lines 99-101**): “Most areas of the northern Bering and Chukchi shelves did not show an early ice retreat or prolonged open water duration from 1996 to 2006 (Supplementary Fig. 2a & 2b).”

7) *Line102, what is “tipping point” in this context?*

Response: The year 2007 was a potential tipping point of ice regime shift. The sentence has been modified (**Lines 101-103**): “The year 2007 was a potential tipping point of ice regime shift, triggered by the transition of a negative Arctic Dipole index to a positive one⁴³.”

8) *Line105, which regions do you refer to here, for “enhanced anomalous oceanic heat flux and amplified ice–albedo feedback”.*

Response: The region in the cited references and main context refer to Pacific Arctic shelf seas. The specific region is added to the sentence (**Lines 103-105**): “The increasingly positive Arctic Dipole index during 2007-2021 drove an enhanced anomalous oceanic heat flux and amplified ice–albedo feedback in the Pacific Arctic shelves^{44,45}.”

9) *Line107, is it “variability” that caused long-term trend?*

Response: Recently, a number of studies have shown that Arctic sea ice declines are related to intense warming in the Arctic, which known as Arctic amplification (Serreze et al., 2011; Previdi et al., 2021; Rantanen et al., 2022). The extreme Arctic temperatures are generally attributed to the atmospheric circulation patterns occurring in the northern hemisphere (Overland and Wang, 2010; Moore et al., 2018). As mentioned in the main context, the variabilities of Arctic Dipole index (Bi et al., 2021; Polyakov et al., 2023) and Arctic Oscillation (Lindsay et al., 2009; Cohen et al., 2014) were associated with the Arctic amplification and increasing ice loss. Besides, warm air masses carried on by extra-tropical cyclones have an increased tendency to invade the whole Arctic Ocean from the Fram Strait on the Atlantic side to the Bering Strait on the Pacific side (Moore, 2016; Kim et al., 2017). Therefore, we think that the long-term decreasing trend of sea ice are related to the patterns of variability in atmospheric circulation. The sentence has been rewritten to be proper (**Lines 106-108**): “The Arctic Amplification⁴⁸ driven by atmospheric circulation patterns caused the seasonal sea ice to decline further over the long term without any sign of recovery⁴⁹.”

- Bi, H. et al. Influences of summertime Arctic Dipole Atmospheric Circulation on sea ice concentration variations in the Pacific sector of the Arctic during different Pacific Decadal Oscillation Phases. *Journal of Climate*, **34**(8): 3003-3019 (2021).
- Cohen J., et al. Recent Arctic amplification and extreme mid-latitude weather. *Nat. Geosci.* **7**, 627–637 (2014).
- Kim, B. M., et al. Major cause of unprecedented Arctic warming in January 2016: critical role of an Atlantic windstorm. *Scientific Reports* **7**, 40051 (2017).
- Lindsay R W., Zhang J., Schweiger A., Steele, M. & Stern, H. Arctic sea ice retreat in 2007 follows thinning trend. *Journal of Climate*, **22**(1): 165-176 (2009).
- Moore, G. W. K. The December 2015 North Pole warming event and the increasing occurrence of such events? *Scientific Reports* **6**(1), 39084 (2016).
- Moore, G. W. K., Schweiger A., Zhang J. & Steele M. Collapse of the 2017 winter Beaufort High: a 587 response to thinning sea ice? *Geophysical Research Letters* **45**, <https://doi.org/10.1002/2017GL076446> (2018).
- Overland, J. E. & Wang, H. Large scale atmospheric circulation changes are associated with the recent loss of Arctic sea ice. *Tellus A* **62**(1), 1-9 (2010).
- Polyakov, I. V. et al. Fluctuating Atlantic inflows modulate Arctic atlantification. *Science* **381**, 972 (2023).
- Previdi, M., Smith, K. L. & Polvani, L. M. Arctic amplification of climate change: a review of underlying mechanisms. *Environmental Research Letters* **16**, 9 (2021).
- Rantanen M., et al. The Arctic has warmed nearly four times faster than the globe since 1979. *Communications Earth and Environment* **3**, 168 (2022).
- Serreze M C. & Barry R G. Processes and impacts of Arctic amplification: A research synthesis. *Global and Planetary Change* **77**, 85-96 (2011)

10) Line 109, -> day of a year

Response: The sentence has been modified (**Lines 109-110**): “day of a year when sea ice concentration drops below 15% for three consecutive days”.

11) Line 119, I do not see what is excluded in (a)

Response: We thank the reviewer for pointing out the confused description here. We have removed the description of study area (water depth) in Fig. 1a (“excluding regions shallower than 15 m or deeper than 500 m”) but rather emphasize this point in the main content (**Lines 78-80**): “Here, we analyze 23,320 shipboard CTD/XCTD profiles and model reanalysis results from 1996 to 2021 to depict the interannual variations and decadal trends of the upper ocean properties on the Pacific Arctic shelf seas (depths of 15-500m).”

12) 126, 144, do you mean “the day when ice is free”?

Response: The definition of ice retreat time in this paper is the day of a year when sea ice concentration drops below 15% for three consecutive days (**Lines 109-110** and Method “**Sea ice concentration**”). When the sea ice concentration is less than 15%, the

grid cell is generally considered as an open sea. The “annual ice-retreat time” in Lines 126 and 144 refers to the ice retreat time in each year.

13) L129, *With no - without (and at other places)*

Response: Those words have been modified (**Lines 108, 129 & 197**).

14) L128, *grid - grid cell (and at other places)*

Response: Those phrase have been modified throughout the manuscript (**Lines 128, 129, 142, 162, 165, 166, 167, 169, 196, 197, 200, 252, 261, Methods and Supplementary**).

15) L137, *what does “it” refer to?*

Response: The “it” in Line 137 refers to the mixed layer depth (MLD). The sentence has been modified (**Lines 137-139**): “Although no significant correlation was found between observed MLD and sea ice concentration, MLD was closely related to the trigger time of sea ice retreat (Fig. 1c).”

16) L145, *remove “a”*

Response: The sentence has been modified (**Lines 146-147**): “Theoretically, the mixed layer would deepen under sufficiently long wind mixing.”

17) L147, *into ... - with sea ice retreat*

Response: The sentence has been modified (**Lines 147-148**): “ice meltwater became less and less influential with sea ice retreat.”

18) L153, *mixed layer – MLD*

Response: The sentence has been modified (**Lines 153-155**): “To assess the impacts of changing sea ice conditions on the upper ocean, we further calculated trends of the mean summer (June to September) MLD using a total of 18,997 quality-controlled hydrographic profiles over two separate periods”.

19) L166, *and then ... - and then the water in the region ...*

Response: The sentence has been modified (**Lines 171-172**): “then the water in the region gradually became saltier (up to 0.08 psu yr⁻¹)”.

20) L170, *why “freshening” trend? I would rather say there is no significant salinity trend*

Response: The sentence has been modified to be more proper (**Lines 175-177**): “In contrast, the mixed layer temperature and salinity on the northern Bering and Chukchi shelves did not show significant trends in both periods.”

21) *L186 models - model results*

Response: The sentence has been modified (**Line 191-192**): “For the high-latitude Chukchi Sea, the summer MLD trends derived from observations were similar to the model results.”

22) *L208 remove two*

Response: The sentence has been modified (**Lines 214-215**): “Significant changes in wind stress mostly occurred in open seas or marginal ice zones for both periods.”

23) *L209, - in the season of June ...*

Response: The sentence has been modified (**Lines 215-217**): “The wind stress from the northern Bering to Chukchi shelves decreased in the season of June to August during 1996-2006.”

24) *L211, why is there growth window for wind stress?*

Response: The original Line 211 was to clarify that wind stress increased in different areas of Pacific Arctic shelves in the season of May to August during 2007-2021. We have rewritten the sentences to eliminate misunderstandings (**Lines 217-220**): “After 2007, the enhancement of wind stress started in May and lasted for nearly 4 months with a positive trend of $1.7 \times 10^{-3} \text{ Pa yr}^{-1}$, which agreed with prior results of climate models^{55,56} and observations⁵⁷.”

25) *L212, - a positive trend*

Response: The sentence has been modified (**Line 218**).

26) L213-214: do you mean that “winds” are strengthened by surface warming? The strengthening in winds is different from a strengthening in wind stress.

Response: Thanks for your question. The magnitude of wind stress at the ice/ocean surface is calculated as the square of the wind speed at 10 m above the sea surface multiplies air density and drag coefficient. We attributed the increase of wind stress to an increase in wind speed of the study region. One of the explanations for the strengthened surface wind speed in Pacific Arctic Ocean is the reduction in surface roughness caused by sea ice loss (Jakobson et al., 2019). Besides, the stronger surface winds are linked to decreased atmospheric stability, which follows from increased temperature of Arctic Amplification (Seo and Yang, 2013; Mioduszewski et al., 2018). In addition, the stronger sea level pressure gradient between the Aleutian Low and the Beaufort High promoted geostrophic winds (Zhang et al., 2016; Huang et al., 2022).

The paragraph has been rewritten to further clarify the points (**Lines 219-224**):

“The strengthened wind stress was related to wind speed-up over the Pacific Arctic shelf under Arctic Amplification⁵⁸, because sea ice loss⁵⁹ and surface warming⁶⁰ decreased ocean surface roughness and atmospheric stability. The stronger pressure gradient between the Aleutian Low and Beaufort High also promoted geostrophic wind speed^{61,62} with the positive Arctic Dipole regime since 2007⁴⁵.”

Huang, Y., Bai, X. & Leng, H. Synoptic-scale variability in the Beaufort High and spring ice opening in the Beaufort Sea. *Frontiers in Marine Science*, **9**, 929209 (2022).

Jakobson, L., Vihma, T. & Jakobson, E. Relationships between sea ice concentration and wind speed over the Arctic Ocean during 1979–2015. *Journal of Climate*, **32**, 7783–7796 (2019).

Mioduszewski, J., Vavrus, S. & Wang, M. Diminishing arctic sea ice promotes stronger surface winds. *Journal of Climate*, **31**(19), 8101–8119 (2018).

Seo, H. & J. Yang. Dynamical response of the Arctic atmospheric boundary layer process to uncertainties in sea-ice concentration. *J. Geophys. Res. Atmos*, **118**, 12383–12402 (2013).

Zhang, J., et al. Mesoscale climatology and variation of surface winds over the Chukchi–Beaufort coastal areas. *Journal of Climate* **29** (8), 2721–2739 (2016).

Overland, J. E., Francis, J. A., Hanna, E. & Wang M. The recent shift in early summer Arctic atmospheric circulation. *Geophysical Research Letters* **39** (19), L19804 (2012).

27) L215-216: *The Arctic Dipole has two active centers, not just “anticyclonic”*

Response: Thank you for pointing out the loose expression on the Arctic Dipole, which has been corrected. The sentence has been rewritten (**Lines 222-224**): “The stronger pressure gradient between the Aleutian Low and Beaufort High promoted geostrophic wind speed^{61,62} with a positive Arctic Dipole regime since 2007⁴⁵.”

28) L217: *You did not illustrate “wind growth” or increase of wind speed in the paper. Wind stress is not only influenced by winds and an increase in wind stress does not prove an increase in wind speed.*

Response: Thank you for reviewer’s suggestion. The original L213-217 sentences within the whole paragraph was to explain the potential causes of increased wind stress after 2007. Based on the calculation of wind stress, we speculated that the strengthened wind stress was related to the increase in wind speed. Several lines of evidence from the literature can support the phenomenon and mechanism of increased wind speed in the Pacific Arctic Ocean. The paragraph has been rewritten to clarify the point (**Lines 219-224**).

29) *Fig S9 does not show a systematic increase in wind stress in June to Sept in the latter period, while the TKE increased more substantially. Why?*

Response: Ocean surface currents driven by the increased wind stress will result in increased levels of turbulent mixing. The increasing exposure and duration of open water to stable surface wind stress will also promote more transfer of momentum from the atmosphere to the ocean (Rainville et al., 2011; Muilwijk et al., 2024). The substantial increased TKE is the combination of reduced sea ice cover and increased wind stress.

The turbulent kinetic energy (TKE) calculated using 1D sea ice-ocean mixed layer model is related to the surface buoyancy flux and the state of mixed layer (see Methods **1D sea ice-ocean mixed layer model**). The differences in salinity and temperature between the mixed layer and the entrainment zone, and the value of MLD affect the TKE. The wind stress is not equal to the momentum that cause turbulent mixing in the water column. Besides, the increased wind stress in May leads to a potential deepening of the mixed layer. There, the MLD value in May has a chain reaction on the determination of TKE in the season of June to September.

In addition, tides, eddies, submesoscale dynamics, and lateral process can also generate internal waves that increase turbulent mixing (Rippeth and Fine, 2022), although those processes are beyond the scope of the present study.

Rainville, L., Lee, C. M. & Woodgate, R. A. Impact of wind-driven mixing in the Arctic Ocean. *Oceanography* **24**(3), 136-145 (2011).

Muilwijk, M., Hattermann, T. Martin T., & Granskog, M. A. Future sea ice weakening amplifies wind-driven trends in surface stress and Arctic Ocean spin-up. *Nature Communications*, 15, 6889 (2024).

Pippeth, T. P. & Fine, E. C. Turbulent mixing in a changing Arctic Ocean. *Oceanography* **35**(3-4), 66-75 (2022).

30) L227, - *their mixtures?*

Response: The sentence has been modified (**Lines 233-235**): “The Chukchi shelf contained complex water masses, including Alaska Coastal Water, ice meltwater, dense winter water, Bering Sea water, and their mixtures in summer⁶⁶.”

31) L229, - *other water ...*

Response: The sentence has been modified (**Lines 235-236**): “Nevertheless, the timing and extent of advected ice meltwater and other water masses varied among regions and years⁶⁷.”

32) L230, *could you explain what variables are 3-month-mean in your analysis? In Method, you mentioned that daily data were used.*

Response: Thanks for your question. We applied a 1D sea ice-ocean mixed layer model to separate and diagnose competing factors for the upper mixed layer evolution, including wind mixing induced turbulent kinetic energy TKE (Fig 4a), surface buoyancy flux (Fig 4b), and sea-ice-derived meltwater (Fig 4c). The linear trends of TKE and surface buoyancy flux are calculated based on the mean value from June to

September, along with MLDs. The sea-ice-derived meltwater trends are determined by the mean value from May to July since the whole region becomes almost ice-free in August. The sentence has been modified (**Line 236-238**): “Linear trends of turbulent kinetic energy and surface buoyancy fluxes were calculated using 3-month-mean to smooth water mass advection and other short time scale processes.”

33) L236-237, *this rather tells that changes in ice meltwater play a role! Or, without the changes in sea ice meltwater, MLD in the latter period would have been different. So it is not just that wind mixing played the role.*

Response: Thank you for your excellent suggestion. We agreed that the reduced sea-ice-derived meltwater indeed played an important role. We emphasize the role of ice meltwater in the abstract and conclusions.

Abstract (Lines 37-39): “We attribute mixed layer deepening to markedly lengthened open water duration, enhanced wind-induced mixing, and ice meltwater reduction.”

Conclusions (Lines 282-286): “In contrast, as the Pacific Arctic shelves shifted to a new normal with extremely low summer sea ice after 2007, the upper mixed layer reversed from shoaling to deepening, suggesting that sea-ice-derived meltwater was less effective in maintaining strong summer stratification.”

34) L241-242, - Linear trends of buoyancy flux associated with ...

Response: Thank you for your suggestion. Figure 3 shows the linear trends of air-sea heat and freshwater flux based on ERA5 forcing data (Methods: Equations 3 and 4). The determination of surface buoyancy flux is different in the open sea and ice cover region. To avoid ambiguity, we have revised the caption of Figure 3 as “Linear trends of air-sea buoyancy fluxes associated with heat and freshwater” (**Line 250**).

35) L264, *for the first period, you did not show “increasing sea-ice-derived meltwater”*

Response: The perennial ice in the Pacific Arctic Ocean steadily decreased and reached its second record-low extent in 2007 (Comiso, 2006; Kwok, 2007). The sea-ice-derived meltwater did not show significant changes in the Bering Sea shelf potentially due to the seasonal sea ice regime has stabilized prior to 1996. The advance of sea ice retreat timing in the Bering Sea shelf after 2007 cause the variability of ice meltwater buoyancy. In the northern Chukchi shelf, the sea-ice-derived meltwater show significant increasing trend near the mean summer sea ice edge (see Fig. 4e and Fig. 1a red dotted line). The increased meltwater buoyancy likely resulted from the melting of perennial ice in the Chukchi shelf. The description of regional increasing sea-ice-derived meltwater in the first period was mentioned in the revised paper (**Lines 242-244**): “The increased ice meltwater buoyancy near the mean summer sea ice edge in northern Chukchi shelf implied the melting of perennial ice before 2006 (Fig. 4e).”

36) L268, - *trend in MLD*

Response: The sentence has been modified (**Line 277**): “a deepening trend in MLDs facilitated by early ice retreat and enhanced wind stress”.

37) L267-269, please rewrite this sentence properly.

Response: The sentence has been rewritten (**Lines 275-278**): “As the pattern of thinner and seasonal sea ice became a new normal after 2007, the exposure of open water to surface wind stress was greatly extended, which resulted in a deepening trend in MLD facilitated by early ice retreat and enhanced wind stress (Fig. 5b & 5d).”

38) L270, - fluxes into the ...

Response: The sentence has been modified (**Lines 278-279**): “Although long-term observations indicated increasing Pacific water inflow with more heat and freshwater fluxes into the region.”

39) L274, due to upper ocean warming?

Response: The sentence has been modified (**Lines 281-282**): “The traditional viewpoints explained that rapid sea ice loss should enhance upper ocean stratification^{27,34} due to upper ocean warming⁶⁹ and/or freshening^{70,71}.”

40) L277, as commented above, the reduction in meltwater could contribute to the increase in MLD.

Response: The main purpose of origin Lines 274-279 was to explain that the effect of wind mixing outcompetes ice meltwater buoyancy and reversed the mixed layer from shoaling to deepening in later period. We agreed that the contribution of reduced ice meltwater in deepening the mixed layer should not be neglected. The paragraph has been rewritten to be more proper (**Lines 282-287**): “In contrast, as the Pacific Arctic shelves shifted to a new normal with extremely low summer sea ice after 2007, the upper mixed layer reversed from shoaling to deepening, suggesting that sea-ice-derived meltwater was less effective in maintaining summertime strong stratification. Meanwhile, the wind stirring during prolonged open water period gradually dominated upper ocean dynamics.”

41) L303, showed – showing

Response: The word has been modified (**Line 315-316**): “Fig. 5 Schematic diagrams showing the dynamic changes of the upper mixed layer under different sea ice regimes.”

42) the description of the 1D model should be improved. The most importantly, it should explain the solution procedure, what is done/solved first, the second step, the third

Response: The 1D sea ice-ocean mixed layer model in the **Methods** has been

rewritten (**Lines 607-676**). Detailed modifications have been made to the model's parametric equations (the determination of entrainment velocity), solution procedures, variable units, etc. The flow chart of 1D sea ice-ocean mixed layer model was added to illustrate the solution steps (**supplementary Figure. 8b**). We also modified “Code availability” and made the code public in ZENODO repository (Lines 758-761): “All Matlab scripts used for data analysis and 1D sea ice-ocean mixed layer model calculation and auxiliary data files are available at the ZENODO (<https://doi.org/10.5281/zenodo.12174955>).”

43) L594, 596, how are T/S_{bottom} and d_T, d_S determined?

Response: The mean temperature and salinity in the mixed layer (MLT, MLS), entrainment zone (T^*, S^*) and sea bottom (T_{bottom}, S_{bottom}) can be obtained from GLORYS2V4 model results. d_T and d_S are the e-folding depths of thermocline and halocline. The determination of d_T and d_S based on the mean temperature and salinity in the mixed layer entrainment zone and sea bottom from GLORYS2V4.

T_{bottom} and S_{bottom} d_T and d_S were previously used to determine the upwelling velocity and entrainment velocity in the origin method. We have modified the calculation of entrainment velocity W_e in the revised paper (**Lines 632-637**). The entrainment velocity W_e is now given by the deepening rate of mixed layer using daily GLORYS2V4 model results and the Ekman pumping/suction velocity. The variables of T_{bottom} and S_{bottom} d_T and d_S are no longer needed.

44) L607, the left and right sides of the equation have different units. Time should be divided on the left side.

Response: Thank you for pointing out this error. we have modified the calculation of entrainment velocity W_e (**Lines 631-636**): For the 1D model, the entrainment velocity W_e is given by the deepening rate of mixed layer $\frac{dh}{dt}$ and the upwelling velocity W as Equation (5):

$$W_e = \frac{dh}{dt} + W \quad (5)$$

45) What is ΔS_{whole} ? How is it determined?

Response: S_{whole} was the salt content of water column, ΔS_{whole} represented the change of salt content in the water column. $\frac{dS_{whole}}{dt}$ represented the change rate of salt content, which was determined by the daily S_{whole} results from GLORYS2V4. ΔS_{whole} was used to the determine the upwelling velocity and entrainment velocity in the origin method. In the modified method, these variables are no longer needed. The entrainment velocity W_e is given by the deepening rate of mixed layer from daily GLORYS2V4 model results and the Ekman pumping/suction velocity.

46) L611, the units on the left and right sides of the equations are different

Response: Thank you for pointing out this written error. We have modified the calculation of entrainment velocity W_e in the revised paper, and this function was deleted.

47) L625, time should be divided to get “rate”

Response: Thanks again for pointing out this error, we have corrected the equation (Line 647):

$$Q_s = (E - P) \times MLS + (MLS - S_{ice}) \times SIC \times \frac{dh_{ice}}{dt} \rho_{ice} / \rho_{water} \quad (7)$$

48) L630, mixed layer – seawater in the mixed layer

Response: The paragraph has been rewritten (Lines 652-653): “Sea ice salinity $S_{ice} = 5 \text{ psu}$ and density $\rho_{ice} = 917 \text{ kg m}^{-3}$ are constants and ρ_{water} is seawater density in the mixed layer.”

Reviewer #2 (Remarks to the Author):

Major comments:

1) 'Pacific Arctic Ocean' is a much larger region than what is studied here and includes the entirety of the Canada Basin (to ~85°N). I suggest replacing with 'Northern Bering and Chukchi Seas' or 'Pacific Arctic Shelf Seas' throughout. This is a problem in the title and lines 49, 95, 196, 262, 301, and likely other places I have missed.

Response: Thanks for your excellent suggestion. We have replaced the 'Pacific Arctic Ocean' to "Pacific Arctic continental shelves" in whole article, both the title and main texts.

Title: Enhanced wind mixing and deepened upper mixed layer in the new normal Pacific Arctic Shelf Seas with low summertime sea ice.

Line 47: The Pacific Arctic Shelf, comprising the northern Bering and the Chukchi seas.

Line 80: upper ocean properties on the Pacific Arctic shelf seas (15-500m).

Line 88: in increasingly ice-free Pacific Arctic shelf seas

Line 94: The sea ice in the Pacific Arctic shelf seas

Line 205: rapidly changing Pacific Arctic shelf seas.

Lines 271-272: the changing upper ocean dynamics in the Pacific Arctic shelf seas.

Line 310: as a whole in the new normal Pacific Arctic shelf seas

2) *Lines 159-171: The updated figure 2 is very useful, and shows that most locations do not correspond to a significant trend for mixed layer depth (e.g., 69 grid boxes out of 76 grid boxes). But Figure 2 is presented focusing on only those few boxes with a significant trend, which I think is misleading. Too much of a concrete conclusion is stated from very inconsistent evidence. A more honest description of these results needs to be included. This does not affect the conclusions that can be drawn from the paper overall, as these are primarily based on GLORYS and ERA5 analysis.*

Response: Thanks for your excellent suggestion that results in much improved presentation of observation results (**Fig. 2**). In the revised paper, we point out the limitation of linear trends calculated by field observation data. Although we acknowledge the imperfection of unevenly distributed CTD/XCTD casts, the consistency of shoaling/deepening trends in sparse grid cells is the starting point of this work. We modified the concrete conclusions of MLD in the main content. However, we thought the description of regional mixed layer salinity and temperature trends is tenable, since the significant changes existed in large number of grid cells (Supplementary Fig. 4 & 5).

The paragraph has been modified as followed (**Lines 160-172**): "Despite the sampling bias and data gaps of field observations in space and time scales, the consistency of MLD shoaling/deepening trends of sparse grid cells from the Bering to Chukchi shelves was notable (Fig. 2a & 2b). MLDs had a similar shoaling tendency in the southeastern Bering shelf during both periods, whereas higher-latitude regions showed exactly opposite trends in specific grid cells (Fig. 2a & 2b). The shoaling trends of MLD on the northern Bering (-2.4 m yr⁻¹ of 2 grid cells, $p < 0.1$; Fig. 2a) and the Chukchi shelves (-1.1 m yr⁻¹ of 5 grid cells, $p < 0.1$) before 2007 agreed with previous studies^{27,28}. However,

after 2007, the mixed layer on the northern Bering and Chukchi shelves showed a remarkable deepening trend (up to 1.3 m yr^{-1} , $p < 0.1$) illustrated by reddish grid cells (Fig. 2b). A freshening trend was observed in some areas of the southeastern Bering shelf (-0.1 psu yr^{-1} , $p < 0.1$) before 2007, and then the water in the region gradually became saltier (up to 0.08 psu yr^{-1}) (Supplementary Fig. 4a & 4c).”

Additional comments:

3) *Line 35: ‘its development’ should be more specific, e.g., its interannual to decadal evolution.*

Response: Thank you for your suggestion. The sentence has been modified (**Lines 33-34**): “Here, we quantified the summertime upper mixed layer and dynamically related its interannual to decadal evolution with sea ice and atmospheric forcing”.

4) *Line 85: ‘another robust mechanism’ and ‘understanding of the phenomenon’ I’m not sure what you’re referring to here.*

Response: In the revised paper, we have clarified the phenomenon and mechanism of traditional understanding (**Lines 83-86**): “This finding demonstrates another robust mechanism and consequence of sea ice loss on the upper ocean dynamics, in addition to a traditional understanding of shoaling mixed layer with more ice meltwater³³⁻³⁵.”

5) *Line 153: suggest changing to ‘trends of the mean summer...’ so that it’s clear how trends are calculated without referring to the methods.*

Response: Thanks for your suggestion. The sentence has been modified (**Lines 153-155**): “we further calculated trends of the mean summer (June to September) MLD using a total of 18,997 quality-controlled hydrographic profiles over two separate periods.”

6) *Line 152-161: It would be useful to state here that the mixed layer shoals in May, if this is true for the entire region. I am getting this impression from other parts of the paper, but it would be good to clarify here. This would clarify that even though the mixed layer is shoaling earlier with time, that does not affect the trend statistics during June – September.*

Response: Thanks for your suggestion. Considering Pacific Arctic shelves span more than 10 latitudinal degrees and ice conditions vary from year to year (Fig 1b), mixed layer may not always shoal in May. The definition of the summertime MLD in the season of June to September followed previous studies for the whole Arctic (Peralta-Ferriz and Woodgate, 2015) and also specific to the Pacific Arctic shelves (Danielson et al., 2020).

Peralta-Ferriz, C., & Woodgate, R. A. Seasonal and interannual variability of pan-arctic surface mixed layer properties from 1979 to 2012 from hydrographic data, and the

dominance of stratification for multiyear mixed layer depth shoaling. *Progress in Oceanography*, **134**, 19-53 (2015).

Danielson, S. L., et al. Manifestation and consequences of warming and altered heat fluxes over the Bering and Chukchi Sea continental shelves. *Deep-Sea Research II*, **177**, 104781 (2020).

7) *Line 182-185: where is the 'inner southeastern Bering shelf'? A box or arrow in Figure 2 would help.*

Response: The inner shelf is the region with water depth less than 50m in the southeastern Bering Sea (Kachel et al., 2002). The description of the inner southeastern Bering shelf water depth is added (Lines **182-183**): "Observation datasets revealed that the inner southeastern Bering shelf (< 50m) had an overall shoaling trend (Fig. 2b)."

Kachel, K. B., et al. Characteristic and variability of the inner front of the southeastern Bering Sea. *Deep-Sea Research II* **49**, 5889-5909 (2002).

8) *Figure 5a-b: What is indicated by the 'E' and downward arrow, Ekman pumping? For winds and Ekman pumping, it would be better to indicate overall values for the summer period somehow as a monthly analysis (Figure S9) shows no increase during e.g., September. Or indicate the monthly variability depicted in Figure S9 in this summary schematic.*

Response: The origin symbol 'E' represents the transfer of momentum from the atmosphere to the ocean by wind stirring, which is quantified by turbulent kinetic energy TKE in the paper. We revised the schematic diagram (Line **314**; Fig **5**) and removed the arrows and symbol 'E' to avoid potential misunderstanding. The spiral line below the mixed layer is used to represent the process of turbulent mixing driven by wind stirring. The density of spirals implied the mean strength of turbulent kinetic energy, which increase in the "new normal" of low sea ice. The drawing of wind stress has also been modified according to your suggestion. We use different colors to represent the regional significant increased (red), regional significant decreased (blue) and no changes (grey) monthly wind stress based on the supplementary Figure. 9.

9) *Figure 5c-d: only a suggestion - in panels c and d, it would be more intuitive to keep the terms (TKE, meltwater etc.) on the same side of the balance in both panels, and change which side is up / down between the two.*

Response: We have modified Figure. 5c & 5d based on your suggestion.

10) *Line 530-541: GLORYS is used to show the mixed layer depth change, but ERA5 for diagnosis of what's causing that change. Does GLORYS use ERA5 forcing?*

Response: According to the quality information document for Global Ocean Reanalysis Products (<http://marine.copernicus.eu/documents/QUID/CMEMS-GLO-QUID-001-025-011-017.pdf>), GLORYS2V4 are forced by ERA-Interim reanalysis products. ERA-Interim is the fourth generation ECMWF atmospheric reanalysis of the

global climate covering the period from January 1979 to August 2019. ERA-Interim has been superseded by the ERA5 reanalysis and users are advised to use ERA5 forcing based on data provider, the Copernicus Service. The description of ERA-Interim and its relationship with ERA5 is mentioned in **Model reanalysis in Methods (Lines 553-555)**: “GLORYS2V4 are forced by ERA-Interim, the fourth generation ECMWF atmospheric reanalysis of the global climate and has been superseded by the ERA5 after year 2019.”

11) **Line 648 – 658**: *Am I correct that for the observations, the mean value from June – September in each year is first calculated, and then a linear trend of those mean values is determined? How does the seasonal cycle bias these results, if observations are in June for one year and September for another? Does this explain why so many grid boxes have an insignificant trend?*

Response: Your understanding is correct. In the Pacific Arctic region, only limited ship-based field observations were conducted. Given changing spatial sampling from year to year, CTD/XCTD profile data were sparse and unevenly distributed. It is difficult to extract meaningful results from field observations without performing some spatial and temporal averaging.

Our research aims to explain the phenomenon of reversed mixed layer from shoaling to deepening between 1996-2006 and 2007-2021. The time span is shorter relative to prior studies of Danielson et al (96 years from 1922-2018) and Peralta-Ferriz and Woodgate (33 years from 1979-2012). The amount of monthly field observations in this paper is insufficient to eliminate the seasonal cycle bias in an accurate way. The maximum sampling years for specific months during 2007-2021 in all $1^{\circ}\times 2^{\circ}$ grid cells of the study region are 11 years, with large spatial and temporal differences. Besides, according to the statistical results of Peralta-Ferriz and Woodgate (2015), the summertime (June to September) mixed layer depths in the Chukchi Sea shelf have a narrow, quasi-normal distribution. Therefore, we used the mean values from June to September in each year to represent the state of summertime mixed layer.

The mixed layer may have significant changes on order days and on small space scales, in addition to a longer seasonal scale (Peralta-Ferriz and Woodgate, 2015). A sudden storm or the advection of ice meltwater in short timescale may affect the field observations. The insignificant trends in the grid cells partly due to the randomness of the observations.

We added descriptions of linear trends from field observation data in **Statistical analysis of linear trends in Methods**, including solution procedures, hypothesis and limitation (**Lines 688-702**): “In this work, we are interested in detecting long-term trends of the upper mixed layer properties from the ship-based observations and model results. In light of the sparse field data coverage in time and location, we bin MLDs derived from the observations of June to September in each $1^{\circ}\times 2^{\circ}$ grid cell to represent the state of the summer upper water column^{27,93}. We only use the grid cell with sufficient sampling years to obtain robust decadal trends. For example, the grid cells on

the shelf containing fewer than 5 years of data in 1996-2006 were excluded from the linear trend analysis. For 2007-2021, each grid cell required at least 6 years of data. According to the statistical results of Peralta-Ferriz and Woodgate (2015)²⁷, the summertime (June to September) mixed layer depths in the Chukchi Sea shelf have a narrow, quasi-normal distribution. Therefore, the long-term trend of MLDs from ship-based observations was estimated as the best linear fit to the mean MLDs of summertime (June to September). It needs to be known that a sudden storm or the advection of ice meltwater in a short timescale may affect the CTD profiling on a daily scale, which causes uncertainty to the results.”

In addition, the linear trends of MLD from GLORYS2V4 is recalculated after removing the seasonal cycle in Fig. 2c & 2d. The linear trends of MLD from GLORYS2V4 model results in Supplementary Fig.6 and Fig.7 was calculated without removing the seasonal cycle for better comparisons with observations.

12) Supplement line 39: typo on ‘2007-201’

Response: The typo has been corrected.